# SELF-IMPROVING VISION-LANGUAGE-ACTION MODELS WITH DATA GENERATION VIA RESIDUAL RL

**Wenli Xiao**[1,2,†], **Haotian Lin**[2,†], **Andy Peng**[1,3], **Haoru Xue**[1,3], **Tairan He**[1,2], **Yuqi Xie**[1],
**Fengyuan Hu**[1], **Jimmy Wu**[1], **Zhengyi Luo**[1], **Linxi "Jim" Fan**[1,‡], **Guanya Shi**[2], **Yuke Zhu**[1,4,‡]

[1]NVIDIA, [2]CMU, [3]UC Berkeley, [4]UT Austin, [‡]GEAR Team Leads, [†]Equal Contributions

## ABSTRACT

Supervised fine-tuning (SFT) has become the de facto post-training strategy for large vision-language-action (VLA) models, but its reliance on costly human demonstrations limits scalability and generalization. We propose Probe, Learn, Distill (`PLD`), a three-stage plug-and-play framework that improves VLAs through residual reinforcement learning (RL) and distribution-aware data collection. In Stage 1 (*specialist acquisition*), we freeze the VLA backbone and train lightweight residual actors via off-policy RL. These specialists take over in states where the base policy fails, thereby probing failure regions of the VLA generalist. In Stage 2 (*data collection*), we employ a hybrid rollout scheme that biases residual interventions toward states frequently visited by the base policy, aligning collected trajectories with the generalist's deployment distribution while capturing recovery behaviors. In Stage 3 (*fine-tuning*), these curated trajectories are distilled back into the generalist with standard SFT, applicable to both flow-matching and autoregressive heads. We evaluate `PLD` across diverse settings: it achieves a near-saturated 99% task success rate on the LIBERO benchmark, delivers over 50% performance gains in SimplerEnv, and demonstrates a 100% success rate on real-world Franka arm and YAM arm dexterous manipulation tasks. We further provide ablations showing that residual policy probing and distribution-aware replay are key to collecting deployment-aligned data that improves VLAs' capabilities on both seen and unseen tasks. Our results demonstrate that RL-generated, policy-aligned data can surpass teleoperation-only demonstrations, offering a scalable path toward self-improving VLA models.

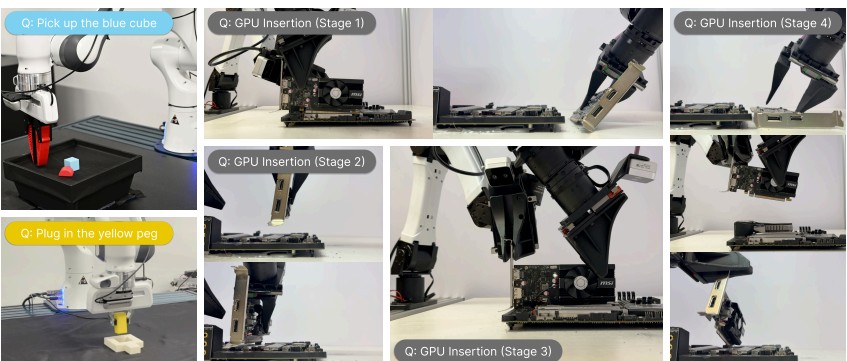

Figure 1: We demonstrate the performance of `PLD` on several real-world challenging manipulation tasks. The robot successfully picks up diverse objects and performs peg insertion on the Franka arm. Additionally, we deploy `PLD` on YAM bi-manual settings, showing that the `PLD` policy continuously performs GPU insertion and unplugging cycles nonstop for 1 hour *without human intervention*. Check videos at https://www.wenlixiao.com/self-improve-VLA-PLD

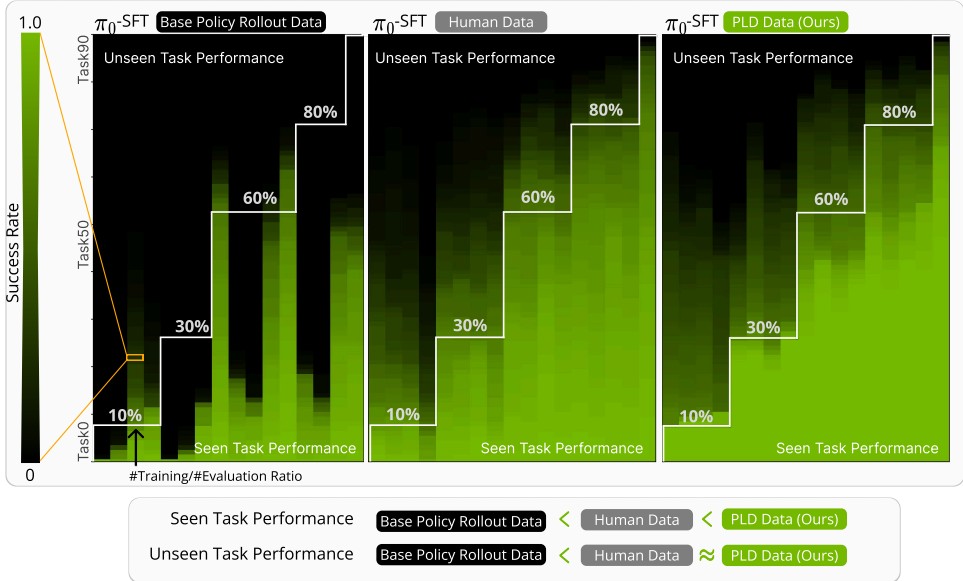

Figure 2: **Synergetic effect of `PLD` data.** We fine-tune $\pi_0$ on subsets of LIBERO-90 with varying **task coverage ratios**, where each ratio (10–80%) indicates the fraction of distinct task instances included in training relative to the full 90-task distribution. For each ratio, we randomly sample 4 disjoint subsets of tasks and report the average results. The x-axis thus represents the degree of task coverage (not the number of trajectories), while the evaluation is always conducted on all 90 tasks. We compare different data formulations: `PLD` data yields the highest in-distribution performance while retaining the cross-task generalization property of high-quality human data. It further enables modest-level zero-shot transfer even when trained on only 10% of tasks (24.4% SR on unseen tasks), whereas the VLA fine-tuned on base-policy rollout data (0-1 REINFORCE) underperforms and fails to generalize. (Success rate numbers are reported in Table 2.)

# 1 INTRODUCTION

Supervised fine-tuning (SFT) has become the standard post-training paradigm for large language models (LLMs): after broad pre-training, models are adapted to downstream applications by training on curated instruction–response pairs, yielding many improvements in language following, safety, and generalization (Ouyang et al., 2022; Team, 2025). Inspired by these successes, the same recipe is now being applied to robot foundation models, particularly vision-language-action (VLA) policies, where large, heterogeneous robotics and vision-language datasets provide the base initialization, and SFT specializes models to specific tasks and embodiments (O'Neill et al., 2024; Team et al., 2024; Kim et al., 2024; 2025; Black et al., 2024; Bjorck et al., 2025). However, transferring this paradigm from language to robotics is a unique challenge. Collecting high-quality robot demonstrations is both costly and labor-intensive, making large-scale datasets much harder to obtain. Even when such data are available, they are often collected through teleoperation pipelines that are *decoupled* from the deployed VLA policy, leaving critical coverage gaps: human operators must manually anticipate and correct failure modes, but their demonstrations rarely reflect the actual distribution of states the policy will encounter at deployment. As a result, while SFT reliably improves performance on the tasks it is trained on, much less is understood about whether these gains transfer to new tasks and environments.

These challenges raise the following question: Can VLA models improve themselves using RL-curated data with minimal human effort? Specifically, can this self-curated training match or surpass fine-tuning on human-expert (oracle) teleoperation data, both in-distribution and out-of-distribution? Our central observation is that data collection should *not* be agnostic to the base policy: the data-collecting policy and the generalist must *interact*, so that exploration leverages the generalist's prior knowledge and collected data remain aligned with its trajectory distribution. A natural way to in-

stantiate this idea is to employ reinforcement learning (RL) to acquire task-specific specialists that guide data collection. However, applying RL in this setting is hindered by two key challenges. Sparse reward signals in language-conditioned manipulation tasks render RL unstable and sample-inefficient. Moreover, training task-specific experts independently from the generalist introduces distributional mismatch, and once these experts converge, their behavior often lacks the diversity needed to provide robust coverage for SFT.

Motivated by these challenges, we introduce `PLD`, a three-stage post-training pipeline. **Stage 1: Online specialist acquisition.** We *freeze* the VLA backbone and train several lightweight *residual* actors for multiple tasks via sample-efficient off-policy RL, enabling them to "take over" the base policy at arbitrary states and achieve above 99% task success. **Stage 2: Automatic data collection.** We propose a hybrid rollout scheme that biases residual takeovers toward states frequently visited by the base model, mitigating distribution shift while capturing recovery behaviors. **Stage 3: Supervised fine-tuning.** The collected data for multiple tasks are distilled back into the base model through SFT, a process agnostic to VLA architectures, supporting both flow-matching and autoregressive action heads (Black et al., 2024; Kim et al., 2024). An overview of our pipeline can be found in Figure 3. With `PLD`, we can efficiently acquire task-specific RL experts through VLA-guided exploration. Consequently, the VLA further improves using the `PLD` data, achieving performance above 99% on the LIBERO benchmark.

This paper makes the following contributions: 1) Autonomous post-training recipe. We propose a post-training pipeline that enables VLA models to improve autonomously without relying on additional oracle demonstrations. Our method achieves near-saturated 99% success rates on the LIBERO benchmark, and delivers over 50% performance gains in SimplerEnv, underscoring both its effectiveness on seen tasks and its ability to generalize to unseen ones. 2) Systematic study of RL-generated data. We analyze the key components of automatic data collection most beneficial for SFT, and conduct extensive experiments in simulation and on real robot hardware to examine how *RL-generated data* influences generalization to unseen tasks. 3) Comprehensive empirical validation. We provide large-scale ablations of our design choices. Besides, we showcase >99% success rate on Franka Arm and YAM arm dexterous manipulation tasks. Achieving continuous GPU insertion and unplugging operating for 1 hour without human intervention, offering potential for data-efficient post-training of robot foundation models.

## 2 PRELIMINARIES

### 2.1 TASK FORMULATION

We study language-conditioned manipulation with *sparse binary rewards* using *Vision–Language–Action* (VLA) models as the base policy class. We assume a partially observed control process with horizon $T$, where an episode terminates and resets on task success with a restricted time limit. After each episode, a reward $r \in \{0, 1\}$ is assigned. Let $g$ denote the language prompt of goal specification, and let $o_t$ denote partial observations comprising robot proprioception (e.g., joint angle) and RGB images. The policy consumes $(o_t, g)$ and outputs a 7-DoF action (6-Dof delta pose and 1-DoF continuous gripper command), which we express as $a_t = D_\phi(h_\theta(o_t, g))$, where $h_\theta$ is a vision–language backbone and $D_\phi$ is an action head. Consistent with recent VLA models, $D_\phi$ is instantiated by one of three common families: (i) a *diffusion* or *flow-based* action head for continuous control (Team et al., 2024; Black et al., 2024), or (ii) a *discrete action tokenizer* for autoregressive decoding (Kim et al., 2024; Pertsch et al., 2025).

### 2.2 SUPERVISED FINE-TUNING

Given a VLA policy and a demonstration data set $\mathcal{D} = \{(o_t, g_t, a_t)\}$ of observations $o_t$, goal specifications $g_t$, and expert actions $a_t$, SFT adapts the policy by maximizing the likelihood of conditional action. Letting $x_t = (o_t, g_t)$, the canonical objective is behavior cloning (BC) loss. In contemporary VLA systems, the loss instantiation depends on the action head architecture. Auto-regressive/token heads (Kim et al., 2024; Pertsch et al., 2025) train with sequence NLL over action tokens $u_{1:K}$:

$$\mathcal{L}_{\text{AR}}(\theta) = - \mathbb{E}_{k \sim [K]} \big[ \log p_\theta \big( u_k \mid u_{<k}, x \big) \big],$$

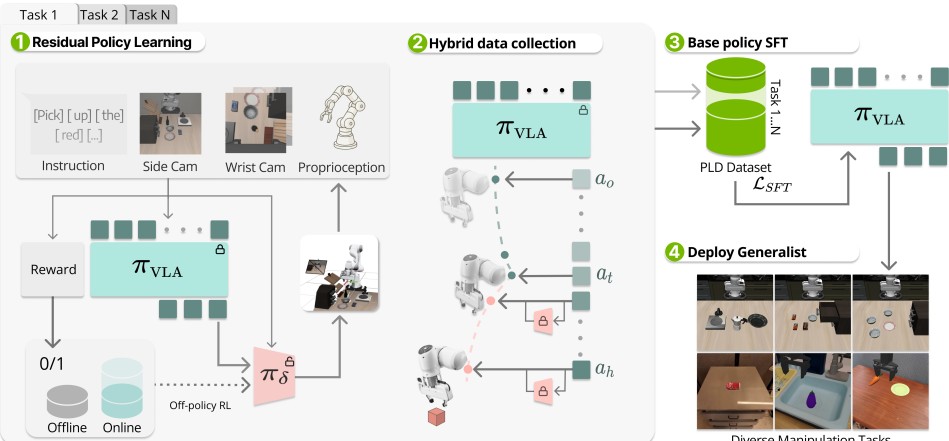

Figure 3: **An overview of `PLD`.** Our pipeline consists of three stages: 1) learning specialist residual policy for each task via online off-policy RL, with efficient exploration guided by a frozen VLA generalist; 2) Automatic generation of hybrid trajectories by having the VLA rollout for the first $t$ steps and let the specialist takeover to generate recovery data; 3) Supervised fine-tuning using collected multi-task `PLD` data; 4) Deploy the fine-tuned generalist to diverse manipulation tasks in zero-shot.

With recent work improving efficiency via action chunking and parallel decoding, and a continuous action parameterization trained by an $\ell_1$ regression objective (Kim et al., 2025). Diffusion heads model a conditional denoising process for actions and train via score-matching MSE:

$$\mathcal{L}_{\text{diff}}(\theta) = \mathbb{E}_{t,\epsilon,(x,a)}\left[\left\|\epsilon - \epsilon_\theta(a_t^{(\text{noisy})}, x, t)\right\|_2^2\right],$$

enabling iterative sampling at inference (Team et al., 2024; Chi et al., 2024). Flow-matching heads learn a continuous velocity field to transport a prior to the action distribution, trained with an $L_2$ flow-matching loss, and are often paired with VLM backbones for semantically grounded control (Black et al., 2024; Intelligence et al., 2025). Across these heads, SFT remains the standard mechanism to specialize a generalist policies to new embodiments and tasks using modest labeled robot data (Kim et al., 2024; 2025).

## 2.3 GOAL-CONDITIONED RL

We model continuous control as an MDP (Bellman, 1957) $\mathcal{M} = (\mathcal{S}, \mathcal{A}, \rho, \rho_0, r, \gamma)$ with state space $\mathcal{S}$, action space $\mathcal{A}$, transition dynamics $\rho(s' \mid s, a)$, initial-state distribution $\rho_0$, reward function $r$, and discount $\gamma \in (0, 1]$. In *goal-conditioned* settings, each task is specified by a goal variable $g \in \mathcal{G}$ drawn from $p(g)$; the reward becomes goal-dependent $r : \mathcal{S} \times \mathcal{A} \times \mathcal{G} \to \mathbb{R}$, and the policy $\pi : \mathcal{S} \times \mathcal{G} \to \Delta(\mathcal{A})$, is written as $\pi(a \mid s, g)$. It is convenient to view GCRL as an augmented MDP on $\mathcal{S} \times \mathcal{G}$ with stationary goals:

$$\tilde{\rho}\big((s', g) \mid (s, g), a\big) = \rho(s' \mid s, a) \cdot \mathbf{1}\{g' = g\}.$$

Under the infinite-horizon setting, the RL objective is

$$J(\pi) = \mathbb{E}_{g \sim p(g)} \, \mathbb{E}_{s_0 \sim \rho_0, \, a_t \sim \pi(\cdot \mid s_t, g), \, s_{t+1} \sim \rho(\cdot \mid s_t, a_t)}\Big[\sum\nolimits_{t=0}^{\infty} \gamma^t \, r(s_t, a_t, g)\Big]. \tag{1}$$

We consider a sparse binary reward setting, i.e., $r(s, a, g) = \mathbf{1}\big[d\big(\phi(s), g\big) \le \varepsilon\big]$ defined via a success predicate over a goal-relevant representation $\phi(s)$, a metric $d$, and tolerance $\varepsilon > 0$.

## 3 METHODS

**Method Overview** We study the synergy between the data produced by our method when a modest *generalist* VLA serves as the prior policy. The premise is that, if we exploit the base policy's

prior correctly, it can both *solve hard tasks quickly* and *explore efficiently*. While recent work explores direct RL fine-tuning of expressive policy class (Mark et al., 2024; Dong et al., 2025b), such a formulation can be resource-intensive even for single-task tuning: e.g., OpenVLA-OFT requires per-GPU memory up to ~62.5 GB for LIBERO training at batch size 8 (Kim et al., 2025). Meanwhile, it remains unclear whether these approaches scale gracefully to multi-task fine-tuning under heterogeneous setups. We therefore opt for a *decoupled* pipeline. We freeze the base policy $\pi_b$ and learn a lightweight residual action policy $\pi_\delta$ with sample-efficient off-policy RL (Gaussian policy parameterization). We then *collect expert data* by letting the residual "take over" after specified steps of "base policy probing". Finally, we *distill* these skills back into the base model via SFT and deploy the generalist on various manipulation tasks. We provide an overview of **PLD** in Figure 3.

## 3.1 DATA EFFICIENT RL VIA POLICY PRIOR WARM-START

Building upon the previous success of sample-efficient RL with prior data (Ball et al., 2023), we consider an off-policy actor-critic framework and maintain two separate buffers for offline and online experience replay. We first fill the offline buffer with successful rollouts $\mathcal{B}_{offline} = \{\tau_1, \tau_2, \dots\}$ from the base policy $\pi_b$. This process serves as an importance sampling to preserve only successful attempts. During training, the offline and online experiences will be replayed symmetrically; for example, mini-batches consist of equal samples from both buffers, ensuring that the value function is constantly trained on high-value state-action pairs.

In practice, we train a task-specific residual action module $\pi_\delta(\cdot|s, a_b)$ conditioned on $a_b \sim \pi_b$. We use $\pi_\delta$ to explore near the base policy behavior, actively searching for more optimal solutions guided by the Q-function. To modulate exploration and avoid deviating drastically from $\pi_b$ during the initial phase, the delta action's magnitude is scaled down to $[-\xi, \xi]$, where $\xi \in [0, 1]$ is tuned by a scheduler. This design choice is two-fold: First, although unable to perfectly generalize to an unseen manipulation task or scenario, the base policy can make reasonable attempts to solve the task, serving as a useful initialization for exploration. Moreover, directly training the expressive foundation policy (e.g., flow action heads) to maximize the Q-value can be extremely difficult (Mark et al., 2024). In contrast, a residual Gaussian policy can be easily trained through any off-the-shelf off-policy RL algorithm.

Alongside $\pi_\delta$, action value function $Q^{\bar{\pi}}$ acquired through policy iteration and TD-learning (Sutton & Barto, 2018) as in Eq. equation 2, where $\bar{\pi}(\cdot|s) = \pi_b(\cdot|s)\pi_\delta(\cdot|s, a_b)$ is the combined policy.

$$Q^{\bar{\pi}}(s_t, \bar{a}_t) \leftarrow r(s, a) + \gamma \mathbb{E}_{s_{t+1} \sim p(\cdot|s_t, \bar{a}_t)}[Q^{\bar{\pi}}_{target}(s_{t+1}, \bar{a}_{t+1})], \ \bar{a} = a_b + a_\delta \qquad (2)$$

To stabilize off-policy learning and mitigate forgetting, we introduce a warm-up stage using solely $\pi_b$ for data collection akin to (Zhou et al., 2024b). Meanwhile, the Q-function is initialized by a conservative objective such as Cal-QL (Nakamoto et al., 2024). Importantly, we do not explicitly enforce behavior constraints to policy loss, such that the resulting expert $\bar{\pi}$ is less influenced by either data quality or base policy performance.

## 3.2 BOOTSTRAPPING RL SPECIALIST FOR SCALABLE DATA GENERATION

We then turn to the question of how to collect demonstration data using RL specialists. Data collected through RL experts is highly optimal, with consistent behavior and nearly no hesitation, demonstrating smooth solutions that finish tasks with a shorter horizon. However, such a narrow distribution of unimodal expert behavior may leave out-of-distribution and failure states underrepresented. Thus, scaling purely expert data may not result in a performance gain, but instead risks the generalist overfitting on these data and harming both robustness and generalization (As discussed in the following section).

To mitigate this issue, we propose a hybrid data collection scheme that incorporates base-policy initialization: We first rollout the base policy for random steps, then let the learned residual RL policy take over, resulting in demonstration trajectories $\tau_{demo} = \{(s_1, a_{b,1}), \dots, (s_{t-1}, a_{b,t-1})\} \cup \{(s_t, a_{b,t} + \bar{a}_t), \dots\}$ that contain the behavior of the expert recovering from a potential suboptimal region. We refer to this procedure as **base policy probing**. Accordingly, we boost the robustness of the RL expert by training the RL expert on an initial state distribution $s_0 \sim p_0^{\pi_b}$ given by random steps of base policy probing. The probing step only serves as state initialization and will not be added to the replay buffer. The details of **PLD** are summarized in Algorithm 1.

Figure 4: **Visualization of Data diversity.** We visualize `PLD` data with different base policy initialization probing horizons. Increasing probing horizon yields longer episodes and greater diversity among successful trials. This broader data support leads to improved fine-tuning performance, which eventually saturates. (As the saturation curve shown in Figure 13).

## 4 EXPERIMENTS

In this section, we systematically evaluate the effectiveness of `PLD`. We first demonstrate the efficiency of `PLD`-RL in solving sparse-reward manipulation tasks, which serves as the cornerstone of our pipeline. Then we focus on study 1) How does the probing mechanism of `PLD` benefit VLA SFT; 2) How does `PLD` data compare with other sources of demonstrations. Finally, we investigate the key factors of our pipeline and how they contribute to improving the performance of VLA.

We consider simulation as a proxy to real-world performance, and evaluate methods across two widely adopted simulation benchmarks, including **LIBERO** (Liu et al., 2023), **SimplerEnv** (Li et al., 2024). LIBERO is a lifelong learning benchmark focused on language-guided manipulation tasks. It comprises 130 tasks grouped into four suites that stress object distribution, spatial arrangement, task goals, and their mixture. SimplerEnv is a robotics manipulation benchmark that aims for high sim-to-real correlation.

In the following sections, we will be mainly analyzing different data sources: `PLD` data $\mathcal{D}^{\text{PLD}}$, Human data $\mathcal{D}^{\text{Human}}$, RL expert data (RL expert rollout w/o base policy probing) $\mathcal{D}^{\text{RL}}$, and base-policy rollout data (Selective successful rollouts, also referred to as "self-bootstrap data") $\mathcal{D}^{\text{Base Policy}}$. Unless stated otherwise, all methods use identical data volume, training budgets, augmentation, and hyperparameters across architectures; the default base policy we used is $\pi_0$ (Black et al., 2024).

### 4.1 EFFECTIVENESS AND EFFICIENCY OF LEARNING RL SPECIALIST

In this section, we seek answers to the following questions: Does `PLD` benefit from both policy guidance and hybrid online learning? We compare state-of-the-art methods that leverage policy priors and data priors: **WSRL** (Zhou et al., 2024b) (offline initialization only); **RLPD** (Ball et al., 2023) (No base policy guidance). For the pre-training stage, we collect a dataset of 50 trajectories per task, containing only the successful trials of the same base policy ($\pi_0$), and using Cal-QL (Nakamoto et al., 2024) as the default pre-training algorithm. Subsequently, we retain these data for methods with online hybrid data replay. We plot the training curve of 250k steps of online interaction, showing mean rollout performance and 95% CIs (confidence levels) across 3 seeds in Figure 5.

`PLD` outperforms baseline methods by a large margin across 8 tasks on LIBERO-90, indicating that `PLD` effectively exploits the VLA policy prior and yields pronounced sample efficiency at low interaction budgets. In terms of asymptotic performance, `PLD` can achieve *over 95% performance* on every task that we report to fine-tune performance (over **120** manipulation tasks). Notably, we observe an initial performance drop for `PLD`. This phenomenon implies the initial phase of exploration, where the residual policy starts to diverge from the base policy and visits potentially suboptimal states. Ablation study on `PLD`-RL's design choice can be found in the Section E.2.

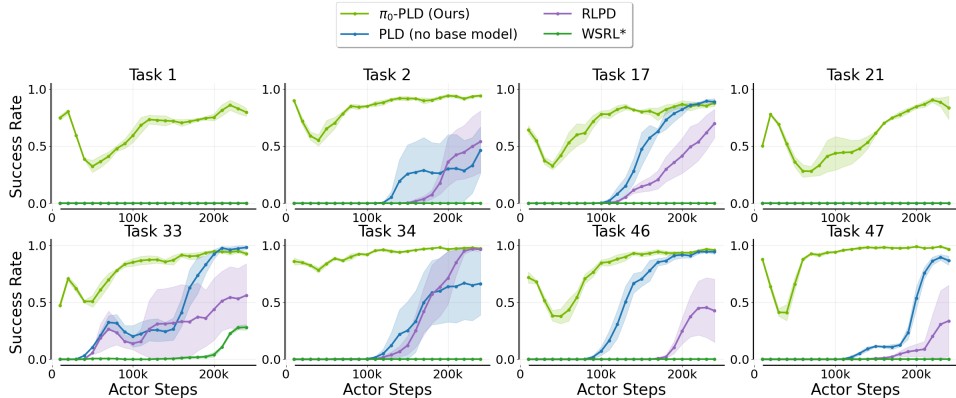

Figure 5: **Benchmarking Sample-Efficient RL Performance.** We compare `PLD` with RL baseline algorithms that either leverage policy prior or data prior. We report mean rollout performance (Average return calculated within a sliding window of 100 episodes) and 95% CIs for 3 seeds across 8 manipulation tasks selected from LIBERO-90.

Table 1: Performance on LIBERO benchmark of VLA models fine-tuned on `PLD` data.

| Model | $\pi_0$ | | | | OpenVLA | | | |
|---|---|---|---|---|---|---|---|---|
| | Spatial | Object | Goal | Avg | Spatial | Object | Goal | Avg |
| Baseline (SFT/OFT) | 95.2 | 97.6 | 87.4 | 93.4 | 92.9 | 99.1 | 83.25 | 91.8 |
| *w/ PLD* | **97.7** | **98.5** | **95.3** | **97.2** | **99.5** | **99.1** | **98.9** | **99.2** |
| $\Delta$ | +2.5 | +0.9 | +7.9 | +3.8 | +6.6 | +0.0 | +15.7 | +7.4 |

## 4.2 IN-DISTRIBUTION PERFORMANCE

In this section, we investigate how effectively the proposed pipeline enhances the performance of the VLA. We evaluate in-distribution fine-tuning on the LIBERO benchmark using three subsets, each consisting of 10 language-conditioned tasks: *LIBERO-Object*, *LIBERO-Spatial*, and *LIBERO-Goal*. We additionally report results on a custom suite that consists of 4 tasks from SimplerEnv. To demonstrate architecture-agnosticism, we instantiate the base VLA with (i) **OpenVLA** (autoregressive action tokens) (Kim et al., 2024) and (ii) $\pi_0$ (flow-matching action head) (Black et al., 2024). Since VLA models are mainly trained on real-world datasets that cannot work out of the box on simulation benchmarks, we leverage their official checkpoints for model fine-tuning on each benchmark as the baseline. At test time, each policy is evaluated on *50 episodes per task*, and we report the mean success rate per suite and average over the benchmark. Table 1 and Table 3 list the performance gain achieved by further applying our method. Across all suites and both architectures, `PLD` data yields consistent absolute gains over human-only SFT while requiring *no additional human demonstrations*. We observe that larger `PLD` datasets monotonically improve in-distribution success and that the distilled generalist notably surpasses the average specialist, indicating effective transfer of task-specific competence into the base VLA.

## 4.3 GENERALIZATION

**Generalization to Unseen tasks** To study the synergetic effect of `PLD` data, we examine whether `PLD` data improves *zero-shot* performance on unseen tasks in the LIBERO benchmark (Liu et al., 2023). Concretely, we fine-tune $\pi_0$ via SFT using data drawn from the disjoint *coverage subsets* of LIBERO-90 in proportions $\{0.1, 0.3, 0.6, 0.8, 1.0\}$; for each coverage level, we randomly sample tasks to form a new subset in distribution and then evaluate all tasks in the suite. We sampled 4 subsets for each coverage level to provide a more unified result. We consider three different data sources: (i) Ours $\mathcal{D}^{PLD}$, (ii) human expert data $\mathcal{D}^{Human}$, and (iii) *self-bootstrapping* data $\mathcal{D}^{\pi_0 \text{ rollout}}$ (SFT on this variant corresponds to 0-1 REINFORCE (Shenfeld et al., 2025)). We visualize the result in Figure 2. Across coverage levels, $\pi_0$ fine-tuned on $\mathcal{D}^{\mathrm{PLD}}$ attains the strongest in-distribution performance and maintains robust zero-shot transfer to unseen tasks; human data-only SFT achieves

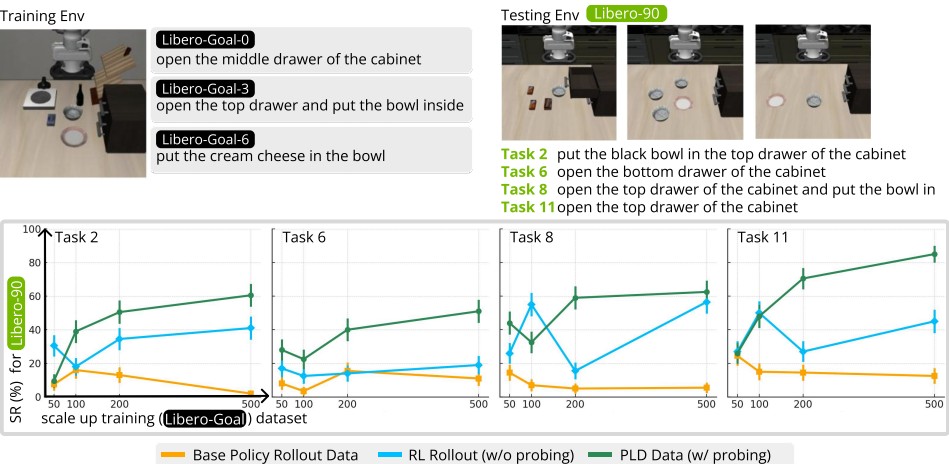

Figure 6: **Few-shot generalization.** Scaling in-distribution (LIBERO-goal) `PLD` yields better few-shot performance on new tasks (LIBERO-90).

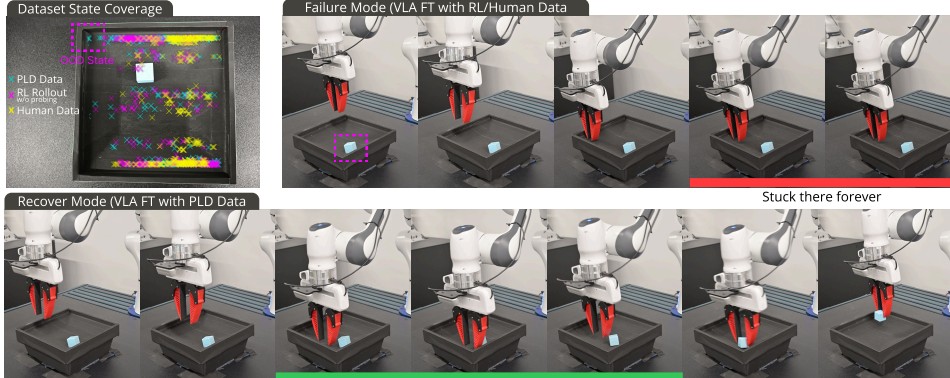

Figure 7: **Visualization of failure mode and recovery behavior in the real-world.**

approximately a similar level of zero-shot generalization at the same training budget but lags on in-distribution tasks; $\pi_0$ self-bootstrapping rollout data underperforms in-distribution and fails to generalize to out-of-distribution tasks.

**Generalization to Out-of-domain** We study *few-shot generalization* for tasks with different goals, layouts, and backgrounds. We first collect `PLD` data of varying scales set on *source* tasks (LIBERO-Goal) and evaluate the fine-tuning performance on *target* tasks (LIBERO-90). Specifically, the VLA is fine-tuned on `PLD` data of source tasks *plus* a small number of oracle demos of target tasks. To analyze transfer by skill family, we select tasks from LIBERO-goal and LIBERO-90 that have high semantic correlation to form a set of source/target tasks. We scaled the size of $|\mathcal{D}^{\mathrm{PLD}}|$ from 50 to 500 trajectories and compared against $\mathcal{D}^{RL}$ and $\mathcal{D}^{BS}$ under the same data and training budget. As shown in Figure 6, we observe monotonic improvements in SFT performance as the data scales from 50 to 500 trajectories.

## 4.4 REAL-WORLD PERFORMANCE

We evaluate our approach on a 7-DoF Franka Emika Panda arm in the real world, considering two sets of canonical manipulation tasks: *pick-and-place* and *peg insertion*, illustrated in Figure 18. Unlike prior works (Luo et al., 2025; Zhou et al., 2024b), we do not restrict task randomization, making real-world reinforcement learning particularly challenging. A more detailed experimental setup is provided in Section G.1.

**Data collection and policy training.** We first collected 200 teleoperated trajectories to perform supervised fine-tuning (SFT) of the base policy $\pi_0$. Using this initialization, we trained $\pi_0$-**PLD** and $\pi_0$-RLPD without human interventions. Both policies reached 100% success on the two tasks within 2 hours of training. We then leveraged the learned expert policies to autonomously collect 200 successful demonstrations each, forming datasets $\mathcal{D}^{\text{PLD}}$ and $\mathcal{D}^{\text{RLPD}}$, which were subsequently used to further SFT $\pi_0$, yielding $+\mathcal{D}^{\text{PLD}}$, $+\mathcal{D}^{\text{Human}}$, and $+\mathcal{D}^{\text{RLPD}}$.

**Performance and failure modes.** Across 30 randomized trials per task, all methods achieved perfect success on peg insertion (30/30), demonstrating robust reactive skills. In cube pick-up, however, $+\mathcal{D}^{\text{RLPD}}$ and $+\mathcal{D}^{\text{Human}}$ succeeded in only 16/30 and 10/30 trials, respectively, while $+\mathcal{D}^{\text{PLD}}$ maintained 30/30. Figure 7 illustrates a typical failure: policies trained on $\mathcal{D}^{\text{RLPD}}$ or $\mathcal{D}^{\text{Human}}$ often pushed the cube into the upper-left corner, where the gripper became stuck. By contrast, $+\mathcal{D}^{\text{PLD}}$ was reliably recovered by repositioning the cube before grasping. Distribution analysis confirms that neither human demonstrations nor RL rollouts visited such corner states, whereas **PLD** explicitly probed the base policy and generated diverse trajectories that captured these cases. This explains its robustness and highlights its potential as a self-improving data flywheel.[1]

**Robustness for long-horizon tasks.** To evaluate the robustness of **PLD** in executing long-horizon and dexterous manipulation tasks, we set up two 6-DoF YAM robot arms developed by I2RT-Robotics (2025). We consider an industrial insertion task—specifically, inserting a micro graphics card into a motherboard. To enable fully autonomous operation without human intervention or resetting, we decompose the task into four stages: Stage 1: Pick up the GPU from the table and insert it into slot 1. Stage 2: Move the GPU from slot 1 to slot 3. Stage 3: Firmly insert the GPU into slot 3. Stage 4: Unplug the GPU from slot 3 and place it back on the table. A reward classifier is trained to govern the state machine that coordinates these stages. After at most 8 hours of training for each subtask and distilling the learned skills into a generalist VLA policy (implementation described in Section F.4), the system can continuously perform the full task loop without human assistance for at least 1 hour. As shown in the video, although the one-shot success rate for each stage is not 100%, the system is capable of recovering from failures, keeping the data flywheel running autonomously.

### 4.5 How does **PLD** work?

We take a deeper look at the underlying reason for **PLD** data's bonus in generalization. As shown in Figure 9, we plot 50 trajectories for each method (task description: "open middle drawer of the middle cabinet"). RL expert provides optimal and concentrated solutions to the task, but lacks diversity and diverges far from the behavior of the base policy, while **PLD** data are clustered near the trials of the base policy and contain various recovery behaviors. Based on empirical observation, we hypothesize that due to the base policy probing, **PLD** data provides a solution that is biased towards the base policy, thus fine-tuning forgets less of the base model's generalizability. This resembles observations in LLM fine-tune (Shenfeld et al., 2025), where the KL-divergence can serve as an indicator of forgetting. Meanwhile, large data coverage also benefits robustness in sequential decision making (Kelly et al., 2019).

## 5 Related Works

### 5.1 Robotics Foundation Models

Following the success of large language models and vision language models (Brown et al., 2020; Touvron et al., 2023; Chen et al., 2022), recent works on robotics foundation models turned to a similar transformer-based architecture with aggressive data scaling. This inspired earlier works in VLAs such as RT-1, RT-2, and OpenVLA, etc. (Brohan et al., 2023b;a; Kim et al., 2024; Xue et al., 2025). Meanwhile, diffusion-based action generation, explored in Chi et al. (2024), takes motivation from generative modeling techniques (Ho et al., 2020), demonstrating smooth and accurate action generation. This has led to more recent VLA architectures to date, such as Octo (Team et al., 2024), OpenVLA-OFT (Kim et al., 2025), GR00T (Bjorck et al., 2025), and the $\pi$-series of models (Black et al., 2024; Intelligence et al., 2025; Pertsch et al., 2025). The VLA training procedure is

---

[1]Generalization results are reported in Section G.

typically analogous to VLM training. First, model weights are initialized from the respective VLM backbones (Kim et al., 2024; Black et al., 2024). Then, the model is supervised with next-token-prediction tasks on diverse pretraining datasets, spanning across multi-modal web data (Intelligence et al., 2025) such as COCO (Chen et al., 2015) and VQAv2 (Goyal et al., 2017), and robotics-specific, cross-embodiment data (Khazatsky et al., 2025; O'Neill et al., 2024). Finally, supervised fine-tuning is conducted on a small set of high-quality teleoperation data collected from the target robot deployment platform performing the target tasks.

## 5.2 SAMPLE-EFFICIENT RL WITH DATA AND POLICY PRIORS

Sample and exploration efficiency have been a long-standing problem in RL, especially in sparse-reward settings. Recent works have explored leveraging offline data to improve sample efficiency. Offline-to-online transfer (Vecerik et al., 2017; Nair et al., 2020; Kostrikov et al., 2021; Nakamoto et al., 2024; Zhou et al., 2024b; Li et al., 2025) considers a two-stage pipeline that first initializes policy or critic using pessimism or constrained objective in offline RL Levine et al. (2020) and follows with an online fine-tuning phase to have new data collected and alleviate distributional shift; Hybrid RL (Song et al., 2022; 2024; Ball et al., 2023) considers online RL with access to an offline dataset. Given expert demonstration, one can either continuously replay this data to ensure high-value state visitation Ball et al. (2023) or to guide exploration Dong et al. (2025a). Data prior can also guide reset-free real-world learning (Walke et al., 2023; Sharma et al., 2023). Another line of work assumes access to policy prior, such as a pre-trained generalist. Ye et al. (2023); Chen et al. (2025); Jülg et al. (2025) leverage foundation policy to guide RL through an auxiliary behavior regularization objective. Action editing is another efficient way to improve upon the policy prior. ResiP (Ankile et al., 2025) considers learning a residual policy through PPO (Schulman et al., 2017), while EXPO (Dong et al., 2025b) considers an off-policy solution and co-trains the base policy during the process. Our work leverages a suboptimal base policy to achieve a non-zero success rate for warm-starting exploration, but does not require access to oracle demos or a human expert for further intervention.

## 5.3 VLA POST-TRAINING

The prevailing large-scale recipe for VLA post-training is to *pretrain* on diverse, heterogeneous robot data and then *fine-tune* on task-specific demonstrations (Zhou et al., 2024a; Black et al., 2024). For example, Black et al. (2024) performs supervised post-training on a carefully curated task-targeted corpus, with per-task coverage ranging from a few to over 100 hours of teleoperation. Because such post-training data are expensive to acquire, the authors note that most diversity must come from the pretraining mixture—underscoring a key limitation of pure SFT: data scarcity and limited coverage at adaptation time. To enable self-improvement, prior work has explored scaling high-quality data via *online RL specialists* (Ball et al., 2023). However, these pipelines often require substantial human intervention and collect data in a large way *agnostic* to the generalist's behavior, restricting scalability. Other lines investigate *on-policy* RL for post-training (Lu et al., 2025; Tan et al., 2025), or optimize *single-task* fine-tuning at the expense of generalization (Chen et al., 2025). Our work jointly targets these limitations by seeking a post-training pipeline that reduces human effort, aligns data collection with the generalist's state distribution, and remains sufficiently sample efficient for real-world systems.

## 6 CONCLUSIONS

We presented `PLD` —a three-stage post-training pipeline that enables VLA models to improve autonomously without relying on additional oracle human demonstrations. `PLD` couples a frozen VLA generalist with lightweight *residual* RL specialists to warm-start exploration and distills curated successes back into the base model with standard SFT. Across large-scale simulation experiments and real-world deployment, `PLD` improves without additional human demonstration, achieving near-saturated ~99% success on LIBERO, >50% gains in SimplerEnv, and robust real-world performance. Ablations identify *residual policy probing* and *distribution-aware replay* as key to stability, sample efficiency, and generalization. We consider `PLD` as a practical step toward autonomous, scalable post-training and a foundation for future work on multi-embodiment transfer, continual on-robot learning, and safety-constrained data collection.

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

# A LARGE LANGUAGE MODEL USAGE STATEMENT

We used the large language model ChatGPT *only* for writing assistance, including grammar correction, wording improvements, and minor stylistic edits on draft text. The model was **not** used for research ideation, data collection, dataset labeling, code generation, experiment design, or analysis. All technical content was authored and verified by the human authors. We accept full responsibility for all content in this paper.

# B EXTENSIVE RELATED WORKS

**Robotics Foundation Models** Following the success of large language models and vision language models (Brown et al., 2020; Touvron et al., 2023; Chen et al., 2022), recent works on robotics foundation models turned to a similar transformer-based architecture with aggressive data scaling. This inspired earlier works in VLAs such as RT-1, RT-2, and OpenVLA, etc. (Brohan et al., 2023b;a; Kim et al., 2024; Xue et al., 2025). Meanwhile, diffusion-based action generation, explored in Chi et al. (2024), takes motivation from generative modeling techniques (Ho et al., 2020), demonstrating smooth and accurate action generation. This has led to more recent VLA architectures to date, such as Octo (Team et al., 2024), OpenVLA-OFT (Kim et al., 2025), GR00T (Bjorck et al., 2025), and the $\pi$-series of models (Black et al., 2024; Intelligence et al., 2025; Pertsch et al., 2025). The VLA training procedure is typically analogous to VLM training. First, model weights are initialized from the respective VLM backbones (Kim et al., 2024; Black et al., 2024). Then, the model is supervised with next-token-prediction tasks on diverse pretraining datasets, spanning across multi-modal web data (Intelligence et al., 2025) such as COCO (Chen et al., 2015) and VQAv2 (Goyal et al., 2017), and robotics-specific, cross-embodiment data (Khazatsky et al., 2025; O'Neill et al., 2024). Finally, supervised fine-tuning is conducted on a small set of high-quality teleoperation data collected from the target robot deployment platform performing the target tasks.

**Sample-efficient RL with Data and Policy priors** Sample and exploration efficiency have been a long-standing problem in RL, especially in sparse-reward settings. Recent works have explored leveraging offline data to improve sample efficiency. Offline-to-online transfer (Vecerik et al., 2017; Nair et al., 2020; Kostrikov et al., 2021; Nakamoto et al., 2024; Zhou et al., 2024b; Li et al., 2025) considers a two-stage pipeline that first initializes policy or critic using pessimism or constrained objective in offline RL Levine et al. (2020) and follows with an online fine-tuning phase to have new data collected and alleviate distributional shift; Hybrid RL (Song et al., 2022; 2024; Ball et al., 2023) considers online RL with access to an offline dataset. Given expert demonstration, one can either continuously replay this data to ensure high-value state visitation Ball et al. (2023) or to guide exploration Dong et al. (2025a). Data prior can also guide reset-free real-world learning (Walke et al., 2023; Sharma et al., 2023). Another line of work assumes access to policy prior, such as a pre-trained generalist. Ye et al. (2023); Chen et al. (2025); Jülg et al. (2025) leverage foundation policy to guide RL through an auxiliary behavior regularization objective. Action editing is another efficient way to improve upon the policy prior. ResiP (Ankile et al., 2025) considers learning a residual policy through PPO (Schulman et al., 2017), while EXPO (Dong et al., 2025b) considers an off-policy solution and co-trains the base policy during the process. Our work leverages a suboptimal base policy to achieve a non-zero success rate for warm-starting exploration, but does not require access to oracle demos or a human expert for further intervention.

**VLA post-training** The prevailing large-scale recipe for VLA post-training is to *pretrain* on diverse, heterogeneous robot data and then *fine-tune* on task-specific demonstrations (Zhou et al., 2024a; Black et al., 2024). For example, Black et al. (2024) performs supervised post-training on a carefully curated task-targeted corpus, with per-task coverage ranging from a few to over 100 hours of teleoperation. Because such post-training data are expensive to acquire, the authors note that most diversity must come from the pretraining mixture—underscoring a key limitation of pure SFT: data scarcity and limited coverage at adaptation time. To enable self-improvement, prior work has explored scaling high-quality data via *online RL specialists* (Ball et al., 2023). However, these pipelines often require substantial human intervention and collect data in a large way *agnostic* to the generalist's behavior, restricting scalability. Other lines investigate *on-policy* RL for post-training (Lu et al., 2025; Tan et al., 2025), or optimize *single-task* fine-tuning at the expense of generalization (Chen et al., 2025). Our work jointly targets these limitations by seeking

a post-training pipeline that reduces human effort, aligns data collection with the generalist's state distribution, and remains sufficiently sample efficient for real-world systems.

**RL with Residual Actor**    Residual actors are lightweight policies that output adjustments on top of the base policy's action. Compared to the expressive base policy, the residual actor reduces optimization complexity, circumventing problems regarding computation inefficiency or unstable training dynamics of large models or flow-based policies (Mark et al., 2024; Tan et al., 2025).

Residual RL has been leveraged to address different challenges in robotics. (Haldar et al., 2023) Consider data-efficient imitation refinement to adapt the base policy across robot morphologies and new object configurations; (Yu et al., 2023) bridges embodiment or modality gaps, such as human-hand to robot-gripper transfer or tactile/vision modality mismatches; (Ankile et al., 2025) scales to long-horizon or precision tasks, augmenting a frozen action chunking policy with a residual policy to address the distribution shifts and the lack of closed-loop corrective control. They include the residual policy as a core component at test time, and the training process is limited to simulation and requires sim2real transfer. (Dong et al., 2025b) leverages a residual actor to address the challenge of fine-tuning expressive policies. The small Gaussian policy can be seamlessly optimized using existing off-policy RL to maximize the Q function. Then expressive base policy can match the edit policy's behavior using the imitation objective. In contrast, one of the core motivations for adopting residual RL in this work is to enhance sample efficiency. Residual policy with controlled scale enables exploration that starts from the base policy's trails. Along with other design choices, our method can be directly adopted for real-world learning on physical robot hardware. Also, we consider augmenting the base policy by distilling expert behavior.

## C ALGORITHM

---

**Algorithm 1** `PLD` with base-policy initialization

---

**Require:** $\pi_b, \pi_\delta, Q_\phi, Q_{\phi'}, \alpha, \gamma, \mathcal{B}_{\text{offline}}, \mathcal{B}_{\text{online}}$

  # **Initialization**
  Collect $n$ successful trajectories of $\pi_b$: $\mathcal{D}_{offline} = \{\tau_1, \tau_2, \ldots \tau_n\}$
  Initialize online buffer $\mathcal{D}_{online} = \varnothing$
  Initialize the critic network $Q_\phi, Q_{\phi'}$ with Cal-QL on $\mathcal{D}_{offline}$
  Randomly initialize residual policy network $\pi_\delta$
  # **RL training**
  Freeze $\pi_b$, denote $\bar{\pi}(\cdot|s)$ as the combined policy (base plus residual)
  **for** each RL step **do**
    **if** collect data **then**
      **if** Warm up step **then**
        base model rollout $a \sim \pi_{base}(\cdot|s)$
      **else**
        sample action $\bar{a} \sim \bar{\pi}(\cdot|s)$
      **end if**
      Environment step: $r, s', done = env.step(\bar{a})$
      Add $(s, a, \mu, r, s')$ to buffer $\mathcal{D}_{online}$.
    **end if**
    Equally sample data from online and offline buffer: $b \sim \mathcal{D}_{online} \cup \mathcal{D}_{offline}$
    Calculate TD target by bootstrapping $\bar{\pi}$
    Update $Q_\phi$ by equation 2
    Update $\pi_\delta$ by maximizing the SAC target
    **Polyak update** $\phi' = \rho\phi' + (1 - \rho)\phi$
  **end for**
  #**Base policy SFT**
  For each task, we collect hybrid behavior dataset $\mathcal{D}_{SFT}$:

$$\pi(s_t) = \begin{cases} a_{base}, & t < T_{base} \\ a_{base} + a_\delta, & t \geq T_{base} \end{cases}$$

  **for** each SFT step **do**
    update $\pi_b$ by BC objective.
  **end for**
  Return $\pi_b$

---

## D MORE RESULTS

Table 2: LIBERO-90 Success rate for $\pi_0$ SFT with different dataset.

| PLD Data | | | Base Policy Rollout Data | | | Human Data | | |
|---|---|---|---|---|---|---|---|---|
| Ratio | Overall SR | Seen/Unseen SR | Ratio | Overall | Seen/Unseen | Ratio | Overall | Seen/Unseen |
| 0.1 | 0.314 | 0.941 0.244 | 0.1 | 0.103 | 0.523 0.056 | 0.1 | 0.272 | 0.796 0.214 |
| 0.3 | 0.470 | 0.968 0.259 | 0.3 | 0.068 | 0.198 0.015 | 0.3 | 0.419 | 0.829 0.240 |
| 0.6 | 0.637 | 0.872 0.283 | 0.6 | 0.328 | 0.506 0.062 | 0.6 | 0.611 | 0.829 0.286 |
| 0.8 | 0.745 | 0.864 0.268 | 0.8 | 0.344 | 0.423 0.031 | 0.8 | 0.694 | 0.803 0.256 |
| 1.0 | 0.871 | 0.871 N/A | 1.0 | 0.488 | 0.488 N/A | 1.0 | 0.815 | 0.815 N/A |

Table 3: Evaluate `PLD` on SimplerEnv

| Model | WidowX Pick Eggplant | WidowX Pick Carrot | Google Open Drawer | Google Coke Can | Avg |
|---|---|---|---|---|---|
| Octo-SFT | 65.5 | 43.3 | 92.5 | 85.7 | 71.8 |
| *w/ ours* | 97.8 | 93.9 | 99.3 | 95.5 | 96.6 |
| Δ | +32.3 | +50.6 | +6.8 | +9.8 | +24.9 |

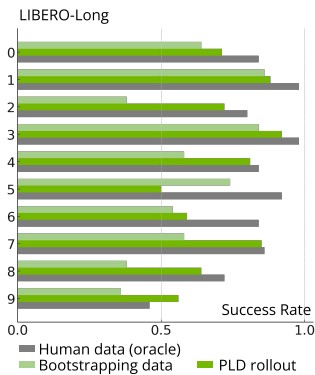

Figure 8: **Short-to-long generalization.** Zero-shot evaluation on LIBERO-10 long horizon tasks.

### D.1 OCTO SFT FOR SIMPLERENV TASKS

In addition to the in-distribution results on LIBERO, we provide results on SimplerEnv in Table 3. Similar to the training and evaluation protocol on LIBERO, we first train task-specific residual RL specialists and fine-tune VLA on `PLD` data for all four tasks.

### D.2 GENERALIZE TO LONG-HORIZON TASK

We assess skill composition on LIBERO-100 by fine-tuning the base VLA on LIBERO-90 (source) and evaluating zero-shot on the held-out LIBERO-10 long-horizon tasks (target). To construct `PLD` data, we first train residual RL specialists independently on each LIBERO-90 task, then aggregate their successful rollouts. As shown in Figure 8, the fine-tuning of the data `PLD` exceeds the tuning of the data from the baseline policy roll-out (self-bootstrapped), but still falls short of the performance achieved with demonstrations by human experts.

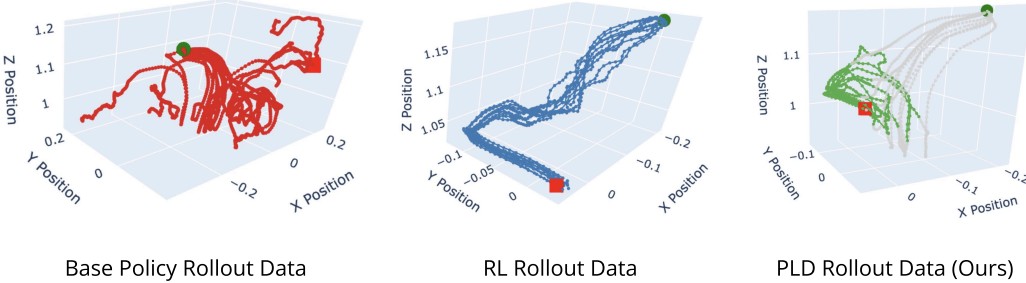

Base Policy Rollout Data    RL Rollout Data    PLD Rollout Data (Ours)

Figure 9: **Visualization of different data sources.** We plot 50 trajectories for each method (task prompt: "open middle drawer of the middle cabinet"). RL expert data is of high quality but lacks diversity and diverges far from base policy behavior, while `PLD` data aligns better with the base policy and contains diverse recovery behavior.

# E  IMPLEMENTATION

## E.1  RL BASELINES

To ensure an apple-to-apple comparison in Section 4.1, we implement these baselines based on the SERL Luo et al. (2024) framework and adapt them to fit in the settings of our study. We provide a detailed explanation of baseline formulation and our implementation.

**RLPD**  RLPD (Ball et al., 2023) proposed a hybrid RL pipeline that leverages offline data to foster learning in challenging sparse reward settings. During training, it equally draws samples from both the online and offline buffer. It also uses LayerNorm to deal with the Q-value blow-ups common when querying OOD actions under a high up-to-date (UTD) ratio. We refer to the implementation in the SERL software for both simulation and real deployment.

**WSRL**  In the original paper (Zhou et al., 2024b), WSRL uses Cal-QL (Nakamoto et al., 2024) to pre-train both the action and critic during the offline phase. For the online phase, it discards offline data and warms up the replay buffer with 50k steps of pre-trained policy rollouts. We did not provide a large dataset that contains diverse behavior as on the D4RL (Fu et al., 2020) benchmark. Rather, we use the same procedure as `PLD` to collect successful trajectories from the base model. We implement WSRL under the SERL framework, as the UTD is no longer fixed to 4. We use the WSRL baseline as an ablation study of the warm-up online exploration using the base policy, and offline data retention through hybrid data replay.

**JSRL**  Jump-start RL (Uchendu et al., 2023) is a meta-algorithm using an existing guide policy to "rolling-in". The key mechanism is to shape the initial-state distribution for the learner: JSRL repeatedly resets episodes from states that the guide visits (a curriculum from easy/near-goal states to harder/far-from-goal states), making difficult tasks learnable with fewer trials. It leverages the guide policy for data collection, without directly imitating its actions. JSRL is agnostic to the underlying RL backbone. In practice, we choose SAC to learn the exploration policy. Since JSRL only leverages the policy prior (VLA policy in practice) to warm-up exploration during online interaction, we use it as an ablation of the hybrid experience replay mechanism.

**Cal-QL**  Calibrated Q-learning (Nakamoto et al., 2024) addresses the underestimation issue of CQL (Kumar et al., 2020), thereby significantly improving fine-tuning performance in the offline-to-online setting. It learns a conservative value function that underestimates the value of OOD actions, while ensuring the values are within a reasonable scale. In practice, it under-bounds the conservative Q function by the value of the behavior policy $\mu$ (policy corresponds to the offline dataset $\mathcal{D}$). The modified Q-learning objective is the following:

$$\min_\theta \alpha \left(\mathbb{E}_{s\sim\mathcal{D},a\sim\pi}[\max(Q_\theta(s,a), V^\mu(s))]\right) - \frac{1}{2}\mathbb{E}_{s,a\sim\mathcal{D}}\left[(Q_\theta(s,a) - \mathcal{B}^\pi\bar{Q}(s,a))^2\right]$$

Where $\bar{Q}$ is the target Q-value function and the second term corresponds to minimizing TD-error Lillicrap et al. (2015).

**Implicit Q-Learning (IQL).**  IQL is an offline RL method that avoids querying the critic function on out-of-distribution queries while still improving over the behavior policy (Kostrikov et al., 2021). The key step is to fit a *state value* $V_\psi$ by *expectile regression* over the actions of the dataset and then bootstrap $Q_\theta$ toward this value. Let $\delta(s,a) = Q_\theta(s,a) - V_\psi(s)$ and define the expectile loss $\mathcal{L}_\eta(\delta) = |\eta - \mathbf{1}\{\delta < 0\}|\,\delta^2$ with $\eta \in (0.5, 1)$. IQL alternates

$$(\text{V}) \quad \min_\psi \mathbb{E}_{(s,a)\sim\mathcal{D}}\big[\mathcal{L}_\eta\big(Q_\theta(s,a) - V_\psi(s)\big)\big], \tag{3}$$

$$(\text{Q}) \quad \min_\theta \mathbb{E}_{(s,a,s')\sim\mathcal{D}}\Big[\big(Q_\theta(s,a) - \big(r + \gamma V_\psi(s')\big)\big)^2\Big], \tag{4}$$

$$(\text{policy}) \quad \max_\phi \mathbb{E}_{(s,a)\sim\mathcal{D}}\Big[\exp\big(\tfrac{Q_\theta(s,a)-V_\psi(s)}{\beta}\big)\,\log\pi_\phi(a\mid s)\Big], \tag{5}$$

which realizes policy improvement without out-of-distribution action queries (the policy step reduces to advantage-weighted regression) (Kostrikov et al., 2021). In our comparison to *Cal-QL* (Nakamoto et al., 2024) as a critic-initialization baseline, we consider a simplified version of

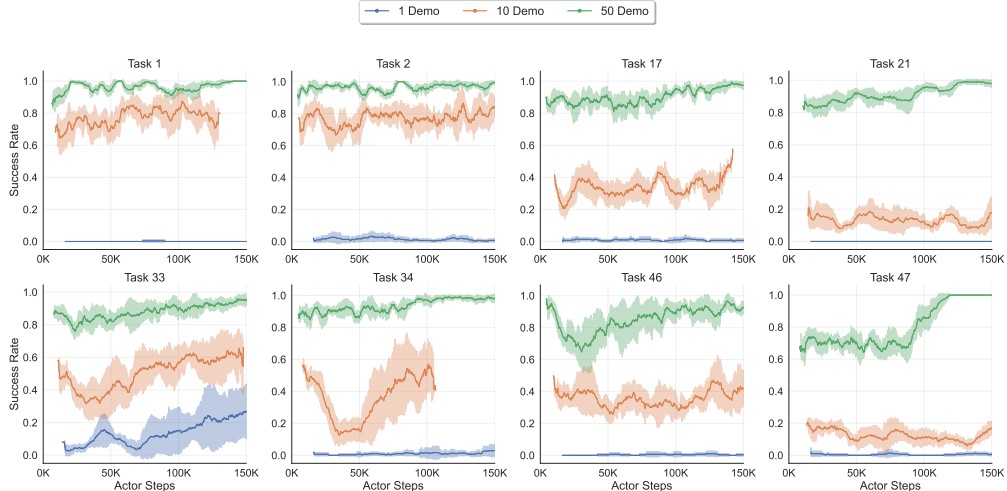

Figure 10: **Base policy ablation.** Mean and 95% CIs of rollout performance across 6 seeds.

IQL that *directly* regresses $Q_\theta$ toward an $n$-step return $R_t = \sum_{i=t}^{t+n-1} \gamma^{i-t} r_i + \gamma^n V_\psi(s_{t+n})$ using expectile regression:

$$\min_\theta \mathbb{E}_{(s_t, a_t) \sim \mathcal{D}} \big[ \mathcal{L}_\eta \big( Q_\theta(s_t, a_t) - R_t \big) \big] .$$

Unless otherwise noted, we set $\eta = 0.7$, a value shown to effectively propagate high-value signals in the IQL paper.

### E.2 DESIGN CHOICES OF `PLD`

In this section, we provide a detailed study of design choices that make `PLD` data efficient and achieve high convergence performance. We evaluate all algorithms on the selected 8 LIBERO-90 tasks.

**Sensitivity to Base Model Performance.** We evaluate how the quality of the base model influences the efficacy of the RL phase in `PLD`. Using a subset of 8 tasks from LIBERO-90, we fine-tune the initial policy $\pi_0$ using 1, 10, and 50 demonstrations, yielding three distinct baselines: $\pi_0^{1\,\text{demo}}$, $\pi_0^{10\,\text{demo}}$, and $\pi_0^{50\,\text{demo}}$. We then apply Phase 1 of `PLD` to each baseline. For the experimental setup, we perform a hyperparameter sweep over action scales $\{0.01, 0.1, 0.5, 1.0\}$ across 6 random seeds, reporting the average performance of the best configuration. As shown in Figure 10, residual RL is highly effective when initialized with a competent policy, boosting success rates to 99% provided the base model achieves at least 80% success. Conversely, the method struggles when the base model is too weak; for example, with $\pi_0^{1\,\text{demo}}$, residual RL fails to converge on 7 out of 8 tasks.

**Sensitivity to the initialization horizon** We choose task 0-9 from LIBERO-90, change the steps we used to initialize the random sample, initiating steps $T_{\text{base}} \sim [0, \alpha T]$ to rollout the base policy. $\alpha \in [0.0, 0.2, 0.4, 0.6, 0.8]$. As $\alpha$ increases, the average episode length of successful trajectories increases, indicating a detour required to correct the suboptimal behavior of the base policy. As demonstrated in Figure 13, performance plateaus at $\alpha = 0.6$ and drops as $\alpha$ increases further. This is consistent with our analysis that SFT benefits from the data diversity.

**Reward shaping** We empirically analyze the impact of naive reward shaping. Specifically, we consider a step-wise *survival cost* as reward bias as in prior works Luo et al. (2024). As shown in Section E.2, adding a slight reward bias has little impact, but it could increase convergence speed in 2 out of 8 tasks; However, A large bias could significantly hinder performance. For the major results reported in the main paper, we do not apply reward shaping.

**Action scale** One core component of residential policies is the scale of exploration. To avoid un-learning results from diverging too far from the base policy, delta actions are usually scaled down

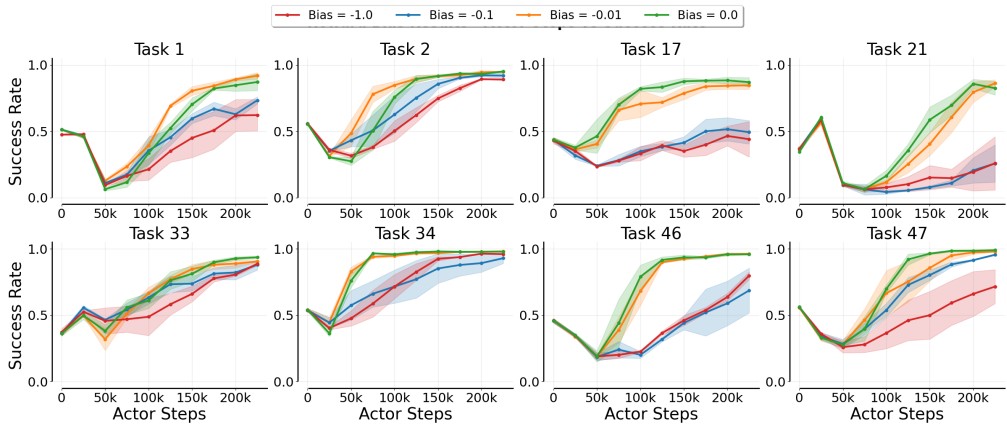

Figure 11: **Reward bias ablation.** Mean and 95% CIs of rollout performance across 3 seeds.

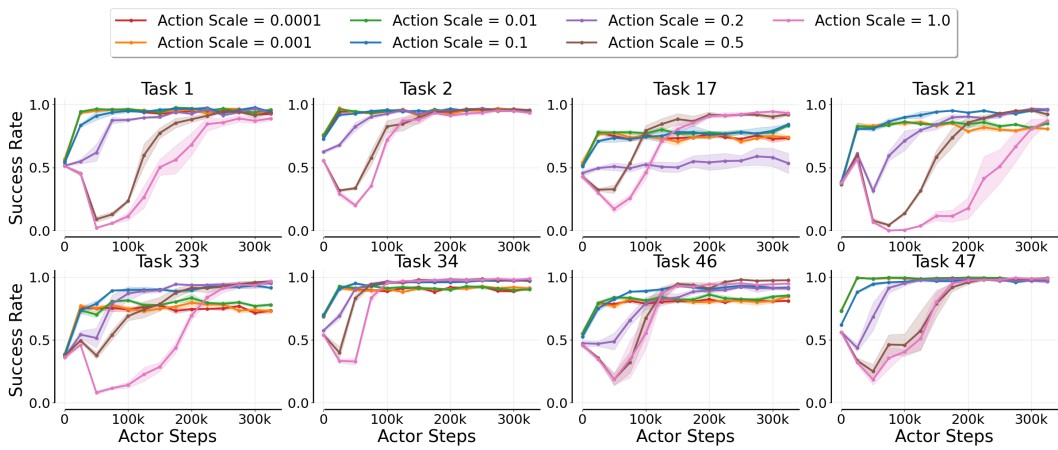

Figure 12: **Action scale ablation** Mean and 95% CIs of rollout performance across 3 seeds.

and bounded within a range of $[-\xi, \xi]$ (Ankile et al., 2025; Dong et al., 2025b). Figure 12 compares different residual action scales. Setting $\xi$ too large at the start can degrade early performance: updates deviate excessively from the base policy, inducing unstable exploration, while a small $\xi$ will lead to insufficient exploration and lower asymptotic performance. We argue that $\xi$ needs to be carefully tuned to enable exploration while minimizing performance drop. For single-arm manipulation, we suggest $\xi = 0.5$ a good choice for LIBERO and $\xi = 0.1$ for SimplerEnv.

**Critic pre-training** While warm-start through pre-training the critic is beneficial to asymptotic performance and prevents initial performance drop, the careful selection of the pre-training method

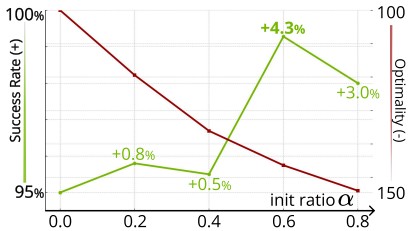

Figure 13: **Ablation of Probing Horizon.** With $\alpha$ measuring the percentage of base policy probing during data collection, we see fine-tune performance plateau at $\alpha = 0.6$. Performance drops monotonically as $\alpha$ increases further.

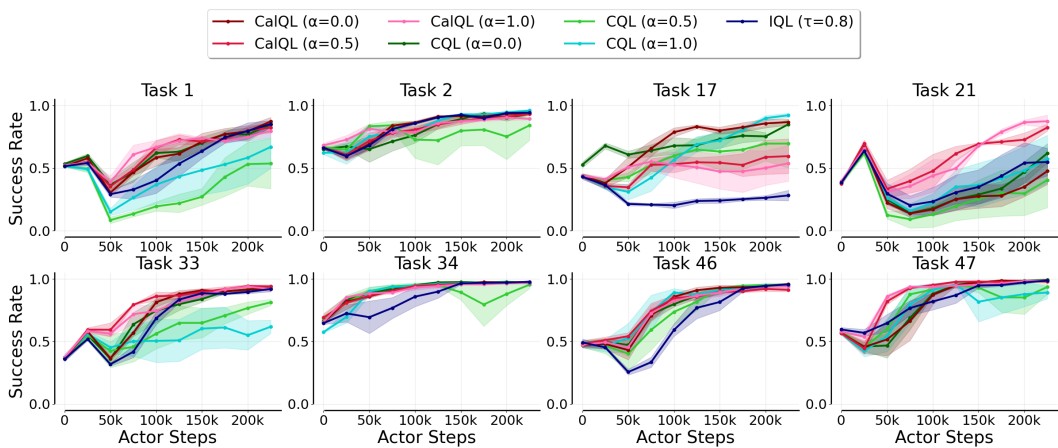

Figure 14: **Offline pre-training ablation.** Mean and 95% CIs of rollout performance across 3 seeds.

could be important as well. We compare using CQL, Cal-QL, and IQL to the pre-training method. We consider using only 50 trajectories of successful trials of the base policy, while the standard offline RL benchmark tends to have far larger data volume (Fu et al., 2020). In Figure 14, online performance using the Cal-QL pre-trained critic is consistently better and is robust to the conservative coefficients $\alpha$. CQL demonstrates the worst performance with a severe forgetting issue, which aligns with the previous study (Nakamoto et al., 2024).

**Update frequency** In the SERL pipeline, data collection and policy learning run asynchronously and periodically exchange network parameters and online data. We ablate the *update frequency*—the number of gradient steps performed by the learner between parameter synchronizations with the data-collection actor—sweeping from 1 to 500. As shown in Figure 15, overall performance is largely insensitive to this hyperparameter, indicating robustness across a wide range of synchronization cadences.

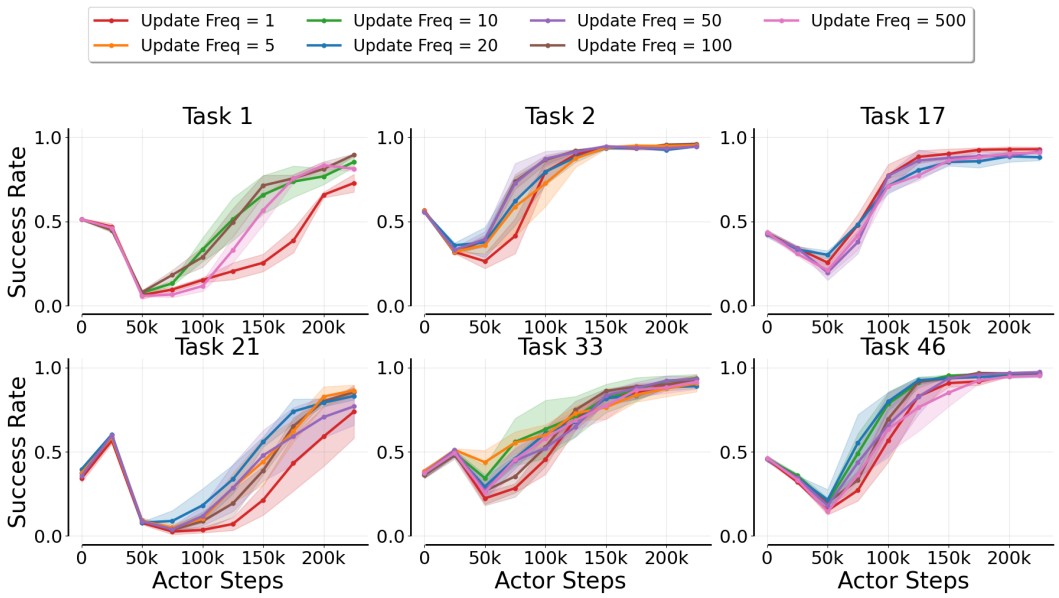

Figure 15: **Update frequency ablation.** Mean and 95% CIs of rollout performance across 3 seeds.

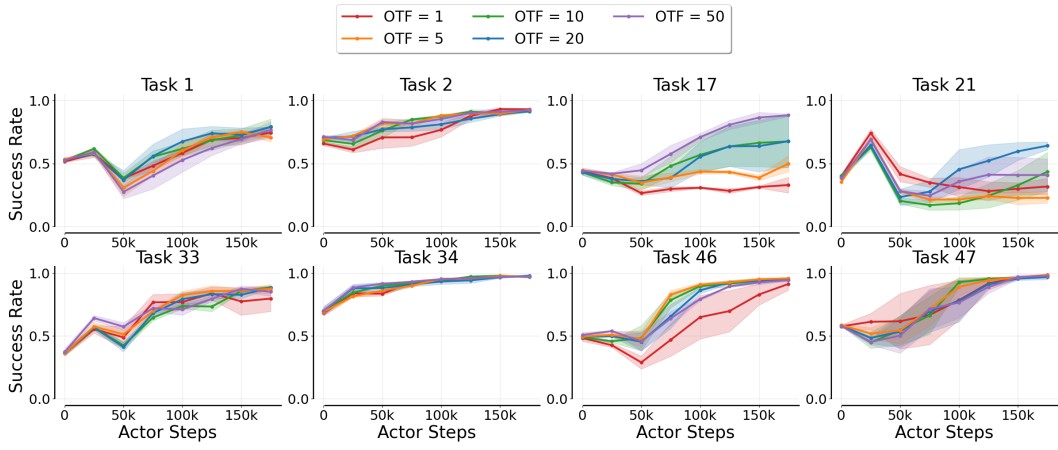

Figure 16: **On-the-fly Policy Ablation.** Mean and 95% CIs of rollout performance across 3 seeds.

**On-the-Fly Policy**   On-the-fly (OTF) policy is introduced in (Dong et al., 2025b) to more effectively maximize the value function. It samples multiple actions and backs up the maximum Q value during TD learning. We adopt OTF to `PLD` while only sampling multiple actions from the residual policy $\pi_\delta$ and conditioned on a fixed base action. We compare different sample sizes in Figure 16. We found that OTF can improve sample efficiency, and a larger sample size ($> 20$) shows significant performance gain. But empirically, the asymptotic performance will eventually be similar. We use OTF= 1 by default.

**JSRL**   We further provide results, including JSRL Uchendu et al. (2023) in Figure 17. We modify the original implementation by opting for a linear scheduler. JSRL demonstrates high data efficiency in general, but could fail to converge on some tasks. While `PLD` can reliably provide solutions for all tasks.

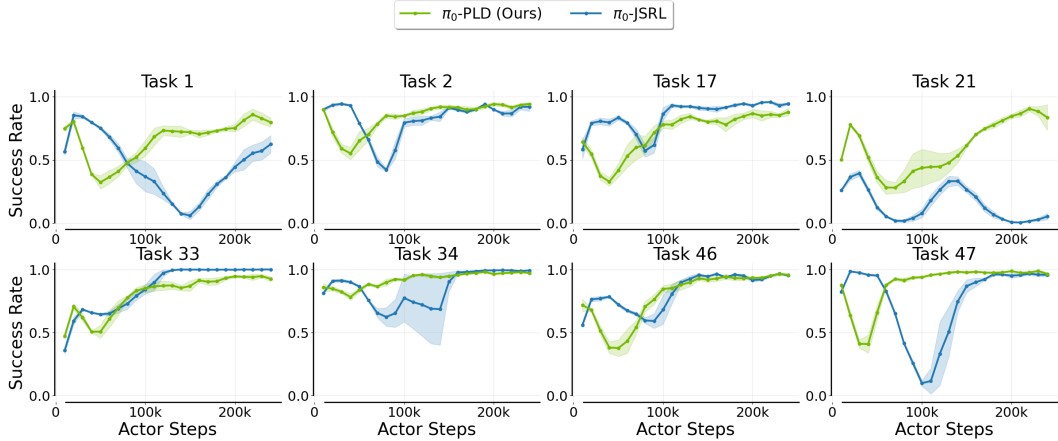

Figure 17: **Compared with JSRL.** Mean and 95% CIs of rollout performance across 3 seeds.

## F  IMPLEMENTATION DETAILS

### F.1  RL ALGORITHM

To ensure apples-to-apples comparisons, all baselines in Section 4.1 and Section E.2 use the same network architecture—an *3-layer MLP Gaussian policy* and *Clipped Double Q-networks (CDQ)* Fujimoto et al. (2018) with *LayerNorm* (Ba et al., 2016). Both actor and critic use a pre-trained ResNetV1-10 encoder to extract visual information. We present a detailed hyperparameter setting in Table 4.

### F.2  COMPUTATIONAL COST ANALYSIS

We analyze the computational resources required for the Residual RL phase (Phase 1) of **PLD**, conducted on NVIDIA L40 GPUs for simulation and an RTX 4090 for real-world experiments. A key advantage of **PLD** is its resource efficiency: since the large VLA base policy remains frozen and we only optimize a lightweight residual MLP, the GPU memory footprint is significantly reduced compared to full fine-tuning. As shown in Table 5, training occupies a peak of only ∼5GB VRAM per task. To further optimize GPU usage, we offload the experience replay buffer to System RAM (up to 100GB per task). This low GPU footprint enables linear scalability for multi-task learning; for example, we successfully parallelized the LIBERO-90 experiment by distributing 90 tasks across a cluster node with 90 L40 GPUs and 10TB of CPU memory.

### F.3  SFT

We employ LoRA (Hu et al., 2021) to fine-tune our VLA base models (OpenVLA and $\pi_0$) efficiently. All SFT experiments are conducted on a node with $8\times$ NVIDIA L40 GPUs. We use a LoRA rank of $r = 32$ and apply the default hyperparameters provided by the respective open-source codebases for both $\pi_0$ and OpenVLA.

### F.4  INFERENCE AND ACTION CHUNKING.

We adopt $\pi_0$ model and Jax (Bradbury et al., 2018) implementation for real-world experiments. To handle high-frequency control in real-world tasks, we utilize temporal action chunking.

- For standard Franka manipulation tasks, we use a chunk size of $k = 20$ and an execution step of $H = 6$ (inferencing every 6 steps to predict the next 20).
- For the YAM arm GPU insertion task, we increase the temporal context, using a chunk size of $k = 26$ and an execution step of $H = 15$.

Table 4: RL hyperparameter settings. We share the same setting across all tasks.

| Hyperparameter | Value |
|---|---|
| **Training** | |
| Batch size | 256 |
| Buffer capacity | 250000 |
| Discount factor ($\gamma$) | 0.99 |
| Gradient clipping norm | 1.0 |
| Learning rate | $3 \times 10^{-4}$ |
| Optimizer | AdamW |
| Reward bias | 0.0 |
| **Residual Policy** | |
| Target entropy | $-\frac{act\_dim}{2}$ |
| Initial temperature ($\tau$) | 1.0 |
| Action scale ($\xi$) | 0.5 |
| **Critic** | |
| Q functions ensemble | 2 |
| Target update rate | 0.005 |
| **Architecture** | |
| Visual Encoder | ResNetv1-10 |
| Hidden layer dimension | 256 |
| Latent space dimension | 256 |
| Q function dropout | 0.0 |
| Activation | Tanh |
| Normalization | LayerNorm |

Table 5: **Computational Cost Analysis.** Resources reported are per single task.

| Setting | Hardware | Peak VRAM | System RAM | Training Time |
|---|---|---|---|---|
| LIBERO (Sim) | NVIDIA L40 | ~5 GB | ~100 GB | 4-6 Hours |
| Real World | NVIDIA RTX 4090 | ~5 GB | ~100 GB | 2-8 Hours |

## G   REAL-WORLD EXPERIMENTS

### G.1   EXPERIMENT SETUP

We deploy `PLD` on a 7-DoF Franka Emika Panda with end-effector delta pose control at 20 Hz. The robot is equipped with one wrist-mounted camera, one side-view camera, and proprioceptive sensing as inputs. For each task, we pretrain an independent binary reward classifier by collecting a small-scale dataset of success and failure states. The model structure follows the setup in (Luo et al., 2025), which uses a pretrained ResNet-10 and a 3-layer MLP model. We ensure the trained classifier using augmented false positive samples until it achieves 99% success rate for each task. Due to the 3D printed desk, we don't need to reset the environment for the pick-cube task. `PLD` performs auto-reset, residual RL training, and SFT automatically without human supervision. For

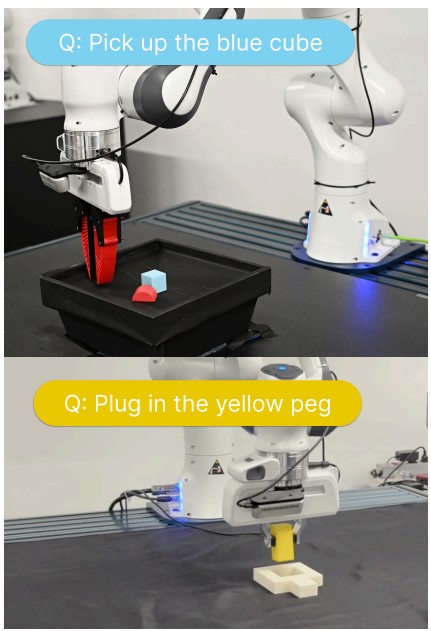

Figure 18: Franka Panda real-world setup for manipulation tasks.

the peg-insertion task (depicted in Figure 19), humans need to randomly move the position of the hole to increase diversity.

## G.2 GENERALIZATION PERFORMANCE

We perform SFT of $\pi_0$ on *Pick Up Blue Cube (Clean Env)* and *Peg Insertion* data, and evaluate the fine-tuned policy on *Pick Up Blue Cube (Cluttered Env)* and *Pick Up Red Cube (Cluttered Env)* tasks. The results in Table 6 show that VLA SFT on `PLD` data achieves better generalization performance compared to human teleoperation data.

Table 6: Comparison of PLD vs Human Data on real-world tasks (success rate).

| SR Dataset | PLD Data | Human Data |
|---|---|---|
| Pick Up Blue cube (cluttered Env) | 28/30 (93.3%) | 12/30 (40.0%) |
| Pick Up Red cube (cluttered Env) | 20/30 (66.7%) | 10/30 (33.3%) |
| Peg Insertion | 30/30 (100.0%) | 30/30 (100.0%) |

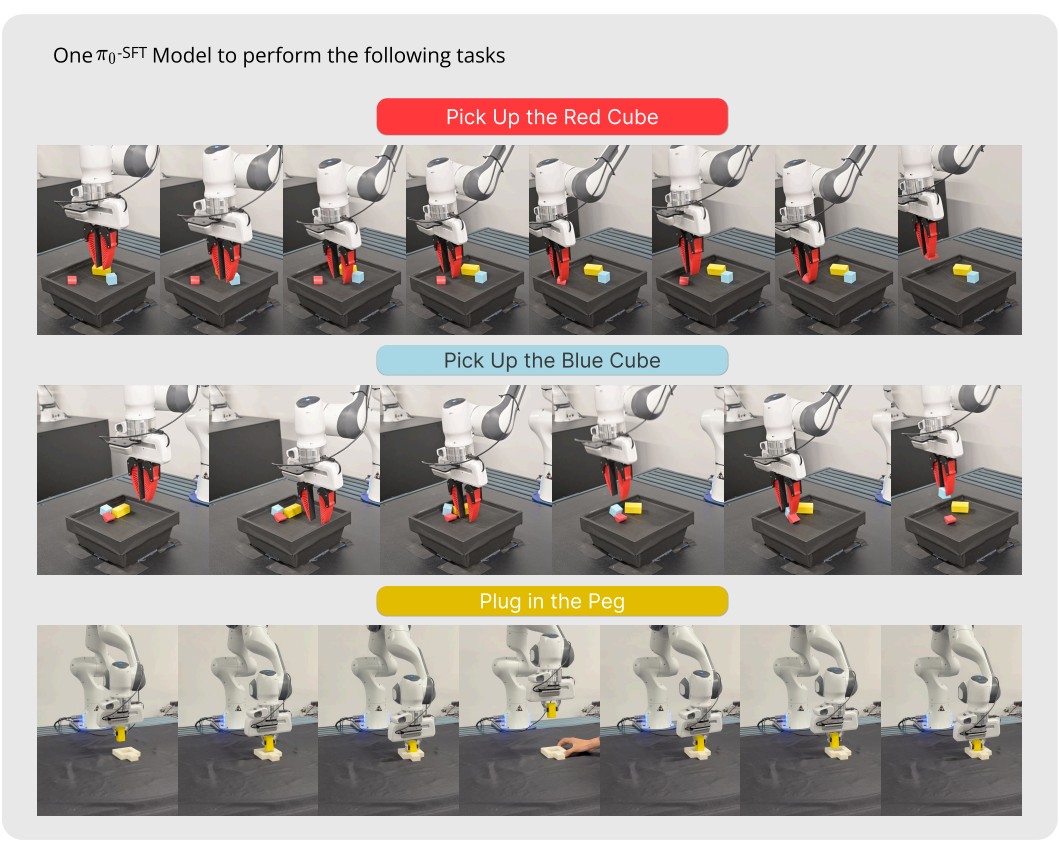

Figure 19: **Real-world Generalization Performance.** We evaluate one model's multi-task performance on three language-conditioned manipulation tasks, including pick-and-place and peg insertion.

