# OpenReview forum: "Self-Improving Vision-Language-Action Models with Data Generation via Residual RL"
_ICLR.cc/2026/Conference — ICLR 2026 Poster_

### Official Review · Reviewer_BWyM · 2025-10-28

**Soundness:** 3
**Presentation:** 3
**Contribution:** 3
**Rating:** 6
**Confidence:** 3

**Summary:**

The paper introduces PLD (Probe–Learn–Distill), a three-stage self-improvement framework for robotic foundation models. PLD enables a VLA model to autonomously refine itself without relying on human demonstrations. It first learns task-specific residual policies via reinforcement learning, then uses a mixed strategy to collect diverse, high-quality data, and finally distills this knowledge back into the base model through supervised fine-tuning. The method achieves near-perfect success rates in both simulation (LIBERO, SimplerEnv) and real-world experiments on the Franka Panda arm, demonstrating strong efficiency, generalization, and practicality.

**Strengths:**

PLD provides a practical approach for VLA models to achieve self-improvement without additional human demonstrations. The data produced by PLD not only aligns with the behavioral distribution of the base model, but also includes recovery behaviors that are typically scarce in human demonstration datasets. Through experiments in both simulated environments and real-world robotic settings, the paper shows that fine-tuning with such self-generated data can effectively enhance the success rate, robustness, and generalization ability of VLA models. This offers a promising solution to the long-standing challenge of high data collection costs in robot learning.

**Weaknesses:**

1. The evaluation focuses mainly on short-horizon, simple manipulation tasks. The approach has not been tested on long-horizon or temporally dependent tasks.
2. The human data only includes successful trajectories, while PLD’s automatically collected data includes failure and recovery cases, which clearly increase diversity. This makes it hard to determine whether the improvement comes from PLD’s three-stage structure or simply from having more varied data. Including ablation studies that remove failure/recovery data from PLD would make the comparison more fair and convincing.

**Questions:**

1. How would PLD handle long-horizon or multi-stage tasks requiring sequential planning?
2. Could the authors provide ablation studies like removing failure/recovery data from PLD?

---

> ### Author Response · Authors · 2025-11-20
>
> We thank the reviewer for the prompt and constructive feedback. Below we address your specific concerns:
>
> > “How would PLD handle long-horizon or multi-stage tasks requiring sequential planning?”
>
> - As detailed in our response to **Reviewer F1ZL (Point 1)**, we added a new real-world Bimanual GPU Assembly experiment. (Please check our demo at [anonymous-pld.github.io](https://anonymous-pld.github.io/))
>
> > “Could the authors provide ablation studies like removing failure/recovery data from PLD?”
>
> - We implemented this suggestion by conducting an ablation where we excluded recovery data from the distillation process, keeping only the successful trajectories generated by the RL specialist.
> - As detailed in **Section 4.4**, the ablated model (without recovery data) exhibits significantly worse performance. Qualitatively, we observe that the policy tends to get stuck in failure modes because it lacks the explicit "correction" signals provided by the recovery data.
> - Furthermore, results in **Figure 6** show that removing recovery data degrades cross-task generalization. These findings validate our core argument: intentionally collecting and distilling recovery data from the base policy's failure states is critical for both robust execution and broader generalization.

---

### Official Review · Reviewer_yzFS · 2025-10-28

**Soundness:** 3
**Presentation:** 3
**Contribution:** 2
**Rating:** 6
**Confidence:** 3

**Summary:**

This paper proposes PLD, a three-stage post-training framework for vision-language-action (VLA) models that reduces dependence on human demonstrations. Stage 1 freezes the VLA and trains lightweight residual RL specialists to improve recovery from failure states. Stage 2 uses a hybrid rollout that starts with the base policy and then hands control to the residual policy, collecting realistic recovery trajectories. Stage 3 distills this data back into the base model via supervised fine-tuning. Experiments on LIBERO, SimplerEnv, and a real Franka arm show strong success rates and generalization, demonstrating a scalable path to improving robotic foundation models with minimal human supervision.

**Strengths:**

- Presents a novel and practical three-stage framework that combines RL and SFT for post-training VLAs without relying on costly human demonstrations.

- Effectively addresses scalability challenges in robot learning by improving data diversity (failure recovery trajectories) and reducing dependence on human demonstration data.

- Provides robust evidence through extensive experiments on LIBERO, SimplerEnv, real-world Franka setups, and systematic ablation studies.

- The paper is well-written and clearly organized, making the methodology and key insights easy to understand.

- Clearly identifies the core challenges preventing direct transfer of LLM post-training successes to VLA models, providing both conceptual and empirical justification for the proposed solution.

**Weaknesses:**

- The real-world experiments focus on short-horizon tabletop manipulation where the base policy is already strong. It remains unclear how PLD performs when initial success rates are low or when recovery requires multi-step planning. Including a few such cases or a discussion of failure modes would better support claims of scalability.

- When applying PLD in the real world, the trade-off between safe operation and sufficient state coverage raises concerns about the method’s ability to handle more challenging, long-horizon, or highly exploratory tasks where recovery behaviors are harder to discover.

- The approach trains multiple task-specific residual RL actors, but the paper does not clarify how many specialists are required as task counts grow, or whether skills can be shared across tasks with related dynamics. Providing guidelines or experiments on task grouping would strengthen the method’s practicality.

**Questions:**

- The real-world results depend on a set of 200 teleoperated trajectories to initialize the base policy. How does the performance of PLD scale with fewer (or noisier) demonstrations, and would the method still succeed if starting from a weaker base policy or no demonstrations at all?

- PLD trains residual experts for multiple tasks, and then distills them back into the base VLA model. Could the authors elaborate on why this distillation step (collect trajectories) is necessary compared to directly deploying the VLA + the residual experts? What specific limitations does PLD address that simple residual policies cannot?

---

> ### Author Response · Authors · 2025-11-20
>
> We thank the Reviewer for the constructive feedback.
>
> > “The real-world experiments focus on short-horizon tabletop manipulation where the base policy is already strong. …” and “... concerns about the method’s ability to handle more challenging, long-horizon, or highly exploratory tasks where recovery behaviors are harder to discover.”
>
> - As detailed in our response to **Reviewer F1ZL (Point 1)**, we added a new real-world Bimanual GPU Assembly experiment. (Please check our demo at [anonymous-pld.github.io](https://anonymous-pld.github.io/))
>
> > “The real-world results depend on a set of 200 teleoperated trajectories to initialize the base policy. How does the performance of PLD scale with fewer (or noisier) demonstrations, and would the method still succeed if starting from a weaker base policy or no demonstrations at all?”
>
> - We conducted an additional experiment in simulation to quantify how base policy quality affects PLD performance (see **Figure 10** in the revision). We initialized the base policy ($\pi_0$) on LIBERO-90 tasks using 1, 10, and 50 demonstrations.
> - We observe a "competence threshold." PLD is highly effective when the base policy achieves a non-trivial success rate (>80%, achieved with 50 demos), boosting performance to near saturation (99%). With fewer demonstrations (1 or 10), where the base policy is weak, the residual policy struggles to learn effectively. This confirms that PLD operates best as a correction mechanism for a rough but semantically capable policy, rather than learning from scratch
>
> >  ”PLD trains residual experts for multiple tasks, and then distills them back into the base VLA model. Could the authors elaborate on why this distillation step (collect trajectories) is necessary compared to directly deploying the VLA + the residual experts? What specific limitations does PLD address that simple residual policies cannot?”
>
> - While deploying the Residual Policy + Base Policy is possible, the distillation step is critical for two reasons: generalization and robustness. Residual policies are typically lightweight MLPs trained on specific tasks; they tend to overfit to the exact training setup and lack visual generalization. By distilling the skills back into the VLA, we can leverage the VLA's large-scale pre-training. As demonstrated in **Section 4.4** and **Appendix G.2**, the distilled VLA policy generalizes to unseen tasks and remains robust to scene changes (e.g., object variations, color), whereas the raw residual policy fails when the environment shifts even slightly.

---

### Official Review · Reviewer_wXQ9 · 2025-10-31

**Soundness:** 3
**Presentation:** 2
**Contribution:** 3
**Rating:** 4
**Confidence:** 5

**Summary:**

A paper for enriching data in finetuning VLA using residual RL. This framework utilize residual RL to generate more diverse and successful trajectory for tuning VLA. To evaluate the performance, this paper conduct comprehensive experiments on the libero benchmark and two real-world experiments.

**Strengths:**

1. Using residual RL to enrich the data for VLA finetuning is promising.

2. The proposed pipeline is well-designed, which includes base policy probing, warm-start, success classifier, and SFT for policy distillation.

3. The experiments in libero benchmark and real-world RL is comprehensive.

**Weaknesses:**

1. Using residual RL to finetune and resolve different gaps is not a pretty novel idea, and this paper didn't disscuss those related work accross different areas in details. For example, residual RL has been used in data efficient learning [1], bridge human2robot embodiment gap [2, 3], real2sim2real transferring for data scaling [4], peg insertion [2, 5], and dexterous manipulations [3, 4]. Disscussing those related work and highlight the contribution might be important.

2. Only binary reward has been used for reward shaping, where it will limit the learning efficiency. Althought it's hard to design dense reward for visual policy or in the real world, there are some related work using VLM [6], generated videos [7], and object-centric representations [8] to do reward shaping.

3. Compared with offline RL or SFT, real-world RL required more human effort to reset and might have unexpected behaviors for some challenging tasks.

4. The paper shows that the control frequency is 20 Hz. For pi0, it can reach high control frequency because of its open-loop action chunking, and its inference frequency is low. Is the residual RL tuning each action of the action chunking and resulting in a close-loop control? However, after SFT, the VLA policy still should be low frequency inference for open-loop action chunking but not close-loop control? For more challenging task, it might require close-loop policy.

5. In the previous real-world paper HIL-SERL [9], they point out that using residual RL to refine a bad BC base policy will result in bad performance. Could you show more insight about this problem? To this end, together with the concern 4, I would be worried about this method's performance in more challenging task, where the policy requires close-loop control, and the VLA base policy might have bad performance.

6. No figure to show the entire pipeline.

[1]. Haldar et al., Teach a Robot to FISH: Versatile Imitation from One Minute of Demonstrations RSS 2023

[2]. Yu et al., MimicTouch: Leveraging Multi-modal Human Tactile Demonstrations for Contact-rich Manipulation, CoRL 2024

[3]. Guzey et al., HuDOR: Bridging the Human to Robot Dexterity Gap through Object-Oriented Rewards, ICRA 2025

[4]. Wan et al., LodeStar Icon LodeStar: Long-horizon Dexterity via Synthetic Data Augmentation from Human Demonstrations, CoRL 2025

[5]. Ankile et al., From Imitation to Refinement – Residual RL for Precise Visual Assembly, ICRA 2025

[6]. Wang et al., RL-VLM-F: Reinforcement Learning from Vision Language Foundation Model Feedback, ICML 2024

[7]. Huang et al., Diffusion Reward: Learning Rewards via Conditional Video Diffusion, ECCV 2024

[8]. Yu et al., GenFlowRL: Shaping Rewards with Generative Object-Centric Flow in Visual Reinforcement Learning, ICCV 2025

[9]. Luo et al., Precise and Dexterous Robotic Manipulation via Human-in-the-Loop Reinforcement Learning, Science Robotics

**Questions:**

1. Residual policy can be different policy architecture other than base policy. In simulator, the residual policy can be simple state-based policy, which might be trained more efficiently. Why still using VLA architecture?

2. For the policy warmstart with expert data, the residual action will always be 0 action. Could you give some more analysis on why this kind of warm start can work? It only warm start the critic or also warm start the residual actor?

3. For those baselins like WSRL, the policies are learned from scratch but not doing residual rl base on the pretrained policy, right? I concern that it might not be a fair comparison.

4. WSRL only uses online data but not offline data after warm starting the Q and show that it's better than off-policy learning. Since this paper utilize similar strategy to warmstart Q using only base policy, why still use offline data?

5. Why don't choose offline residual RL [2] but online RL?

6. I might not an expert in VLM tuning. Since this method still need to do SFT for tuning the VLA, why it requires less computation resource than doing online RL for tuning the VLA directly?

[1]. Xu et al., Compliant Residual DAgger: Improving Real-World Contact-Rich Manipulation with Human Corrections, NeurIPS 2025

---

> ### Author Response · Authors · 2025-11-20
>
> We thank the reviewer for the time and effort in reviewing our paper! We would like to clarify our contribution and design choices for the proposed pipeline:
>
> > “Is the residual RL tuning each action of the action chunking and resulting in a close-loop control? … For more challenging task, it might require close-loop policy.” and “I would be worried about this method's performance in more challenging task”
>
> - We appreciate this insightful consideration. The distilled VLA indeed utilizes action chunking; however, it effectively learned the reactive capabilities of the residual policy. Although the inference within a chunk is open-loop, the model learns from the closed-loop residual expert's corrections during training. This allows the VLA to generate robust action chunks that implicitly account for errors.
> - To demonstrate performance in more challenging tasks, we conduct additional real-world experiments and showcase it can perform precise manipulation tasks like GPU insertion (check our website: [anonymous-pld.github.io](https://anonymous-pld.github.io/)).
> - We acknowledge that closed-loop control is needed for highly dynamic tasks, though not the major focus of this paper, that’s an important direction for our future study.
>
> > ”Using residual RL to finetune and resolve different gaps is not a pretty novel idea, and this paper didn't discuss those related work across different areas in details. … Discussing those related work and highlighting the contribution might be important.”
>
> - We thank the reviewer for pointing out these relevant works. We have significantly expanded **Appendix B. RL with Residual Actor** to provide a detailed comparison with the broader literature on residual RL.
> - While we acknowledge that residual RL is an established concept, we distinguish our work by its application and objective.
> Unlike prior works that often treat the residual policy as a inference-time component, our focus is on a post-training recipe for VLA self-evolution. We use residual RL as a approach to curate high-quality data with minimal human effort, which is then distilled back into the base model. We therefore introduce specific design choices to make this viable on physical hardware with sparse binary rewards. Beyond standard residual formulations, we incorporate: Self-bootstrapped Warm Start: Initializing the actor/critic to jump-start learning; State Distribution Shaping: Using base policy rollouts to guide exploration. Empirically, these contributions allow us to achieve high sample efficiency in the real world.
>
> > ”In the previous real-world paper HIL-SERL, they point out that using residual RL to refine a bad BC base policy will result in bad performance. Could you show more insight about this problem?”
>
> - We thank the reviewer for this insightful connection to HIL-SERL. We agree with the observation that residual RL struggles when the base policy is fundamentally incapable. However, we find that the architecture and the quality of the base policy are the deciding factors.
> - Upon inspecting the [HIL-SERL codebase](https://github.com/rail-berkeley/hil-serl/blob/main/examples/train_bc.py), we noted their base policy is a standard MLP trained with L2 loss, lacking action chunking. This setup is often less expressive and struggles with temporal consistency on challenging tasks. In contrast, we utilize VLA base policies, which benefit from large-scale pre-training. Our base policies possess strong semantic understanding that can often "roughly" complete the task but may fail at specific steps requiring high-precision manipulation.
> - Based on our understanding, Residual RL works best as a "correction mechanism" rather than a replacement. The residual policy learns to take over specifically at critical bottlenecks while letting the base policy execute the majority of the trajectory. To quantify this, we conducted additional sensitivity experiments [see **Figure 10** in the revision]. We found a clear performance threshold: Residual RL is highly effective when the base policy provides a reasonable initialization. Specifically, our results show that if the base policy achieves a non-trivial success rate ($>80\%$), Residual RL consistently boosts performance to near saturation (99%). However, consistent with HIL-SERL's findings, if the base model is too weak (e.g., effectively random or failing early), the residual policy struggles to compensate.

---

> > ### Author Response · Authors · 2025-11-20
> >
> > > “In simulation, the residual policy can be a simple state-based policy, which might be trained more efficiently. Why still using VLA architecture?”
> >
> > - We clarify that we do not use a VLA for the residual policy. As detailed in **Appendix F.1**, the residual agent is a lightweight MLP trained on top of the frozen VLA base.
> > - We use visual observations instead of ground-truth states in simulation to ensure strictly realistic conditions. Our objective is a self-improving pipeline for the real world, where privileged state access is unavailable. By aligning the simulation observation space with the real robot (vision + proprioception), we ensure our method is robust and transferable to physical hardware.
> >
> > > “For the policy warmstart with expert data, the residual action will always be 0 action. Could you give some more analysis on why this kind of warm start can work? It only warm start the critic or also warm start the residual actor?”
> >
> > - We only warm-start the critic using Cal-QL. While the residual action signal is zero in the offline phase, the critic evaluates the joint policy \hat{\pi} and takes in the action that is actually executed (formulation in **section 3.1**).  Thus the critic warm-start works in the same way as in existing offline-to-online pipelines [1].
> >
> > [1] Nakamoto, Mitsuhiko, et al. "Cal-ql: Calibrated offline rl pre-training for efficient online fine-tuning." Advances in Neural Information Processing Systems 36
> >
> > > “For those baselines like WSRL, the policies are learned from scratch but not doing residual rl base on the pretrained policy, right? I concern that it might not be a fair comparison.”
> >
> > - Thank you for raising this point. In our implementation of WSRL, although the method does not use residual RL, both the actor and critic are warm-started using offline data collected from VLA rollouts via Cal-QL. This aligns with the intent of the original WSRL paper, which relies on offline datasets for effective initialization.
> > - However, WSRL does not retain offline data during online RL. Its performance is therefore highly sensitive to the quality and coverage of the initial dataset. In the WSRL paper, the authors rely on D4RL datasets, which combine optimal and failure trajectories, but constructing such balanced datasets requires substantial domain-specific engineering. There is no general or scalable recipe for producing these datasets for new tasks.
> > - We fully acknowledge that WSRL could improve with additional dataset engineering. To address this concern, we also compare PLD against JSRL (**Figure 17**), another method that explicitly leverages a pretrained policy for guided exploration. PLD outperforms JSRL on 6 out of 8 tasks. These comparisons serve as ablations to our design choices (e.g., usage of residual policy, over-sampling).
> > [1] Uchendu et al., “Jump-Start Reinforcement Learning”, ICML 2023.
> >
> > > “WSRL only uses online data but not offline data after warm-starting the Q and show that it's better than off-policy learning. Since this paper utilizes a similar strategy to warmstart Q using only base policy, why still use offline data?”
> >
> > - There are two major reasons. 1) Offline dataset under our setting only contains successful episodes sampled by the base policy, which can already be considered as warm start data that pre-fill the buffer in WSRL. 2) We adopt the over sampling technique similar to RLPD that leverages prior data for fast progress while continuously updating with fresh data, so the policy remains current and adapts. This greatly improves sample efficiency.

---

> ### Author Response · Authors · 2025-11-20
>
> > "Why choose online RL over offline residual RL?"
>
> - We choose online RL over offline residual RL because our objective is to minimize dependence on human-curated datasets and allow the VLA to improve from its own interactions. Offline residual RL necessarily relies on the quality and coverage of the offline dataset, and its performance is bottlenecked when the data lack sufficient corrective behavior or task diversity. In contrast, online RL enables controlled exploration and on-the-fly refinement of the residual policy, allowing the system to acquire behaviors that may be absent or underrepresented in the offline data. Importantly, offline learning (SFT and offline-RL warm starts) still forms the foundation of our pipeline, and we view iterative offline RL—periodically collecting small amounts of real-world data—as a highly promising complementary direction for scalable VLA post-training.
>
> > Why does it require fewer computational resources than doing online RL for tuning the VLA directly?
>
> - Directly applying online RL to fine-tune a VLA model is computationally heavy because it requires frequent gradient updates to the large VLA backbone and substantial parallelization to support multi-task learning in simulation. In contrast, PLD keeps the VLA backbone frozen and uses it only for inference. The learning burden is shifted to lightweight residual actors, which dramatically reduces optimization cost. Moreover, our approach is policy-agnostic: the residual actor isolates the learning signal and avoids the optimization instability and computational inefficiency often encountered when performing RL on large models or flow-based policies [1].
>
> [1] Mark, Max Sobol, et al. “Policy Agnostic RL: Offline RL and Online RL Fine-Tuning of Any Class and Backbone.” arXiv:2412.06685, 2024.
>
> > “Only binary reward has been used for reward shaping, where it will limit the learning efficiency. Although it's hard to design dense reward for visual policy or in the real world, there are some related work using VLM, generated videos, and object-centric representations to do reward shaping.”
>
> - We thank the reviewer for the suggestion. For this work, we focus on the binary reward setting to ensure real-world scalability and minimize per-task engineering. We agree that incorporating VLM-based rewards is a valuable direction for future research.
>
> > "No figure to show the entire pipeline."
>
> - We thank the reviewer for the suggestion. To address this, we have added Figure 3 in the revised paper, which provides a comprehensive visual overview of the entire PLD pipeline.

---

> ### Comment · Reviewer_wXQ9 · 2025-11-21
> **Insightful Response and good Results**
>
> Thanks for your well-structural and insightful rebuttal. Most of my questions and concerns has been answer, more figures have been added, and the new real-world experiment is impressive. To this end, I still have one concern and two small questions. If they can be answered, I would like to increase my score.
>
> 1. It's really impressive if VLA can be used for highly-dynamic USB insertion task, so I go through this part carefully. However, I found that you mentioned you "distilling the learned skills into a single BC base policy", but it's not shown as an VLA policy. Is it a closed-loop BC policy or a pre-trained VLA model? If it's a closed-loop BC policy, I think it's not a good idea to put it as a major contribution in this manuscript (shown in the teaser and the last paragraph of intro), because the title shows that it's "Self-Improving ***Vision-Language-Action Models***".
>
> 2. For the WSRL baseline, since it has been warm-started by the expert data, why the performance is zero at the beginning but not as good as PLD?
>
> 3. I would like to more clearly state my question about off-policy buffer I want to ask. In this paper WSRL, they show that they only use expert data for warm-start ,without using them for online learning because they claim that it will ***restrict the exploration diversity***. Since you use similar warm-start strategy, why do you choose to add replay buffer?

---

> ### Author Response · Authors · 2025-11-21
>
> > … you mentioned you "distilling the learned skills into a single BC base policy", but it's not shown as an VLA policy. Is it a closed-loop BC policy or a pre-trained VLA model? …".
>
> - We thank the reviewer for acknowledging the performance of the USB insertion task. We confirm that the final policy is **indeed a fine-tuned VLA (specifically, $\pi_0$ ), not a simple close-loop BC policy**. The confusion arise from our use of the term "BC base policy" in the draft. We intended this to describe the learning objective (behavioral cloning/supervised fine-tuning) used during the distillation phase, not the model architecture.
> - To support high-dynamic tasks, we utilize the temporal action chunking of VLA. For the GPU insertion task specifically, we apply chunk size of 26 and an execution horizon of 15 steps. Besides, we follow the openpi JAX implementation for fast inference, which allows the VLA to handle the precision and dynamics required for the task.
> - We have revised **Section 4.4 Robustness for long-horizon tasks** and **Appendix F.4** to explicitly state the hyperparameters and the specific VLA architectures used.
>
> > "For the WSRL baseline, since it has been warm-started by the expert data, why is the performance zero at the beginning but not as good as PLD?"
>
> - Poor performance of the initial policy is largely due to the insufficiency of the offline dataset. In the original WSRL paper, even for the “small” dataset in their real-world experiment, the dataset is collected by saving the replay buffer during an online RL run, containing a mixed quality data of 17K transitions (refer to **section 5.7** in [1]). In comparison, the offline dataset in our case is limited to successful trials of the base policies with 50 episodes per task, which is relatively narrow for the challenging manipulation task with sparse binary reward. The evidence also shows that a good policy can not be effectively extracted from an offline dataset at this scale using offline RL objectives (**equation A.4** in [2]), an observation that aligns well with [3].
>
> [1] Zhou, Zhiyuan, et al. "Efficient online reinforcement learning fine-tuning need not retain offline data."
>
> [2] Nakamoto, Mitsuhiko, et al. "Cal-ql: Calibrated offline rl pre-training for efficient online fine-tuning." Advances in Neural Information Processing Systems 36
>
> [3] Park, Seohong, et al. "Is value learning really the main bottleneck in offline RL?." Advances in Neural Information Processing Systems 37
>
> > "I would like to more clearly state my question about the off-policy buffer I want to ask. In this paper WSRL, they show that they only use expert data for warm-start, without using them for online learning because they claim that it will restrict the exploration diversity. Since you use a similar warm-start strategy, why do you choose to add a replay buffer?"
>
> - As explained in the last question, we think WSRL’s claims can not be trivially transferred to our case due to distinct data composition. **The warm start procedure in PLD-RL is different from WSRL** that we only have the critic pre-trained but keep the residual policy random initialized (to initialize the critic with reasonable scale for high value state-action pairs). As we assume a base policy of decent performance, the residual actor can slightly branch out from base-policy’s behavior for better solutions. The offline data replay buffer’s benefit is straightforward. By adapting offline buffer over-sampling, the learner’s visitation of successful states is greatly increased.
>
> It has been an insightful discussion. Should there be any unresolved questions or suggestions for additional experiments, we would be more than happy to discuss them further.

---

> ### Comment · Reviewer_wXQ9 · 2025-11-21
>
> Thanks for your response and explanation, and all of my questions have been resolved. I would like to increase my score and support this paper accordingly

---

> > ### Author Response · Authors · 2025-11-28
> >
> > We sincerely appreciate your thoughtful follow-up and positive evaluation, and we are grateful for your willingness to increase your score and support the paper. Thank you again for your constructive feedback!

---

### Official Review · Reviewer_F1ZL · 2025-11-01

**Soundness:** 3
**Presentation:** 4
**Contribution:** 3
**Rating:** 8
**Confidence:** 3

**Summary:**

This paper proposes a three-stage self-improving framework named PLD (Probe, Learn, Distill) to provide high-quality data to finetune Vision-Language-Action (VLA) models. Its core idea is to freeze the VLA base model and train lightweight residual reinforcement learning (RL) policies to probe the failure regions of the base model; then adopt a hybrid data collection scheme of to align residual interventions with the deployment distribution of the base policy; and finally distill the collected high-quality trajectories back into the base model via SFT. Experiments validate the effectiveness of PLD on the LIBERO and SimplerEnv simulation benchmarks as well as real-world robotic tasks.

**Strengths:**

1. This paper addresses two core pain points of VLA models in SFT: "data dependence" and "distribution mismatch". PLD combines residual RL with distribution-aware data collection. It avoids the high resource consumption of directly fine-tuning large VLA models and solves the poor generalization issue of pure RL expert data, presenting a novel and logically consistent approach.
2. The effectiveness of PLD is well demonstrated through multi-environment (LIBERO, SimplerEnv, real robots), multi-model ($\pi_0$, OpenVLA), and multi-baseline comparisons. Additionally, PLD demonstrates better generalization ability than human data in zero-shot and few-shot experimental tasks.
3. Quantitative analyses are conducted on the impact of key parameters, such as the action scale of the residual policy, the probing ratio of the hybrid rollout strategy, and the pre-training method for the Q-network. This provides reusable parameter selection guidelines for subsequent research.

**Weaknesses:**

1.  Direct comparisons with recent residual RL methods (e.g., EXPO[1], ResiP[2]) are missing. It is recommended to supplement comparative experiments or discussions on this aspect.
2. Current tasks (e.g., object grasping, peg insertion) are relatively simple. It is suggested to validate the method on more complex long-horizon tasks such as cloth folding.
3. The paper mentions that PLD trains lightweight residual policies but does not discuss costs such as training time and GPU memory usage. It is recommended to supplement a computational cost table to clarify PLD’s advantages in "resource efficiency", especially its scalability in multi-task training scenarios.

[1] Dong P, Li Q, Sadigh D, et al. Expo: Stable reinforcement learning with expressive policies[J]. arXiv preprint, 2025.

[2] Ankile L, Simeonov A, Shenfeld I, et al. From imitation to refinement-residual rl for precise assembly[C]//ICRA, 2025.

**Questions:**

The paper states that PLD supports multi-task training. In multi-task scenarios, are residual policies trained independently for each task, or do they share partial parameters to improve efficiency? If trained independently, will PLD’s storage costs (e.g., parameters of residual policies for each task) increase significantly as the number of tasks grows? Are there any parameter sharing or model compression schemes available?

---

> ### Author Response · Authors · 2025-11-20
>
> We thank the reviewer for the positive and constructive feedback! Below, we address your questions and concerns.
>
> > “Current tasks (e.g., object grasping, peg insertion) are relatively simple. It is suggested to validate the method on more complex long-horizon tasks such as cloth folding.”
>
>  - We conduct an additional real-world experiment for a multi-stage precise GPU insertion task to showcase the capability of our method.
>
> - **Setting**: we consider inserting a GPU into a motherboard using bimanual YAM arms. To enable fully autonomous operation without human intervention or resetting, we decompose the task into four stages. Stage 1) Pick up the GPU from the table and insert it into slot 1. Stage 2) Move the GPU from slot 1 to slot 3. Stage 3) Firmly insert the GPU into slot 3. Stage 4) Unplug the GPU from slot 3 and place it back on the table. We trained residual RL experts for each subtask and distilled the skills into a single generalist policy.
> - **Results:** the system can continuously perform the full task loop without human assistance for **at least 1 hour** (please check our video at [anonymous-pld.github.io](https://anonymous-pld.github.io/))
>
> > “Direct comparisons with recent residual RL methods (e.g., EXPO[1], ResiP[2]) are missing. It is recommended to supplement comparative experiments or discussions on this aspect.”
>
> - Thanks for the suggestion. We have added a detailed comparison with prior works, including EXPO and ResiP, in **Appendix B**. Our approach distinguishes itself from these methods in two key aspects: deployment setting and primary motivation. Most prior works (e.g., ResiP) rely on simulation for training and require Sim2Real transfer, or address specific morphology/modality gaps. They often retain the residual policy as a permanent component at test time. In contrast, PLD is designed specifically for real-world RL. By using a residual policy with a controlled scale, we enable safe exploration anchored to the base policy's existing trajectories, allowing us to learn directly in the real world without relying on simulators. While methods like EXPO also distill residual behaviors back into a base policy, their primary focus is addressing the optimization instability of fine-tuning large expressive policies. Our core motivation is sample efficiency. We treat the residual policy as a tool to efficiently gather "correction" data in the real world. Once this high quality data is collected via the residual exploration, we distill it to improve the VLA itself, ensuring the final model remains a standalone, generalist policy without the overhead of a residual adapter.
>
> > “In multi-task scenarios, are residual policies trained independently for each task, or do they share partial parameters to improve efficiency?”
>
> - All residual policies share a fixed, pre-trained ResNet visual encoder. Only the lightweight MLP actor heads are trained independently for each task. Crucially, these residual actors are strictly used for data collection/exploration. Once the exploration phase is complete, their skills are distilled into the single generalist VLA base policy. Therefore, at test time, there is zero additional inference cost or parameter overhead, as the residual modules are discarded.
>
> > "The paper mentions that PLD trains lightweight residual policies but does not discuss costs such as training time and GPU memory usage. It is recommended to supplement a computational cost table to clarify PLD’s advantages in "resource efficiency", especially its scalability in multi-task training scenarios."
>
> - Thanks for the suggestion. We have added a detailed computational cost analysis in **Appendix F.2** and a new summary table (**Table 5**). As detailed in the new section, PLD is highly memory-efficient. Because we freeze the large VLA base policy and only optimize the lightweight residual MLP, the training process occupies a peak of only $\sim$5GB VRAM per task. This allows the method to run easily on consumer-grade hardware (e.g., RTX 4090). We also discuss multi-task scalability. The resource usage scales linearly with the number of tasks. In our experiments, we successfully parallelized the training of all 90 LIBERO tasks simultaneously on a cloud cluster.

---

### Author Response · Authors · 2025-12-03
**Summary of our rebuttal**

We thank the Area Chair for reviewing our paper. This paper proposes PLD (Probe, Learn, Distill), a framework that enables VLA models to self-improve with minimal human supervision via residual RL and distribution-aware data collection. By probing failure regions and distilling recovery behaviors, PLD achieves 99% success on LIBERO and robust real-world performance across two different hardware platforms, offering a scalable path to overcome the data-dependence bottleneck in robot learning.

**Reviewer Consensus** All reviewers acknowledged the significance and effectiveness of our framework. They commended the method for addressing the core limitations of VLA post-training: "data dependence" and "distribution mismatch" [*F1ZL*, *wXQ9*], the soundness of the three-stage design [*F1ZL*, *yzFS*], and the strong empirical results across simulation and real-world benchmarks [*F1ZL*, *wXQ9*, *BWyM*].

**Addressing Key Concerns** We have systematically addressed the reviewers' constructive feedback.

- [*F1ZL, wXQ9, yzFS, BWyM*] Additional Long-Horizon Real-World Task:

  A shared concern was whether PLD scales to complex, multi-stage tasks. We conducted a new real-world Bimanual GPU Assembly experiment. This task requires high-precision insertion and sequential planning. We demonstrate that the distilled VLA policy can perform the full loop autonomously for over an hour. (Video available on [anonymous-pld.github.io](https://anonymous-pld.github.io)).

- [*F1ZL, wXQ9*] Clarification & Differentiation from Prior Residual RL Works:

  Reviewers requested a clearer distinction between PLD and existing residual RL methods (e.g., EXPO, ResiP). We have significantly expanded **Appendix B** to provide a detailed qualitative discussion and differentiation. We clarify that unlike prior works which use residual policies as permanent inference modules, PLD uniquely employs residual RL strictly for data curation to improve the generalist VLA.

- [*BWyM, yzFS*] Ablations on Data & Warm-Starts:
  - Recovery Data: We highlighted our ablation results (**Figure 6**) demonstrating that excluding recovery data significantly degrades performance and generalization, directly addressing BWyM's query on data diversity.
  - Base Policy Quality: We conducted an additional experiment to analyze the "competence threshold" (**Figure 10**), showing PLD works best when the base policy provides a non-trivial initialization, addressing yzFS and wXQ9.

- [*F1ZL, wXQ9*] Computational Cost & Pipeline Figure:

  We added a computational cost analysis (**Table 5, Appendix F.2**) showing PLD is memory-efficient (5GB VRAM) and scalable. We also added a comprehensive pipeline visualization (**Figure 3**).

**Rebuttal Timeline & Status**

- Nov 20: We submitted detailed responses and new experimental results to all reviewers.
- Nov 20: wXQ9 (Score: 4 → Support) explicitly stated that their concerns were resolved, the new experiment is "impressive," and they will increase their score to support the paper.
- Pending: We have provided the requested long-horizon experiments and clarifications to F1ZL (Score: 8), yzFS (Score: 6), and BWyM (Score: 6) and look forward to their final assessment.

---

### Meta-Review · Area_Chair_DaWg · 2026-01-08

**Summary:**

This paper proposes PLD, namely Probe, Learn, Distill, a framework that enables VLA models to self-improve with **minimal** human supervision via residual RL and distribution-aware data collection. By probing failure regions and distilling recovery behaviors, PLD achieves 99% success on LIBERO and robust real-world performance.

This work offers a scalable path to overcome the data-dependence bottleneck in robot learning, which is a crucial problem to be addressed.

**Reviewer Concerns:**

There are four reviews with preliminary rating as 6, 4, 8, 6

The concerns are majorly centered on:

- Additional Long-Horizon Real-World Task
- Clarification & Differentiation from Prior Residual RL Works
- Ablations on Data & Warm-Starts
- Computational Cost & Pipeline Figure

**Reviewer Scores:**

Authors did a good job addressing all the concerns raised by reviewers. The only negative preliminary score is responded by the reviewer as well, showing the willingness to change from negative to positive.

AC read the paper, rebuttal and review comments; and believe this is a solid work towards robot learnig in context of overcoming the data dependance challenge. Good rebuttal!

---

### Decision · Program_Chairs · 2026-01-26

Accept (Poster)